# Morphological changes in the tracheal system associated with light organs of the firefly *Photinus pyralis* (Coleoptera: Lampyridae) across life stages

**Kristin N. Dunn**[1]*, **Steven R. Davis**[1,2], **Hollister W. Herhold**[2], **Kathrin F. Stanger-Hall**[3], **Seth M. Bybee**[4], **Marc A. Branham**[1]

**1** Department of Entomology and Nematology, University of Florida, Gainesville, Florida, United States of America, **2** Division of Invertebrate Zoology, American Museum of Natural History, New York, New York, United States of America, **3** Department of Plant Biology, University of Georgia, Athens, Georgia, United States of America, **4** Department of Biology, Monte L. Bean Life Science Museum, Brigham Young University, Provo, Utah, United States of America

* KristinDunn@ufl.edu

**Data Availability Statement:** The data underlying the results presented in this study are available

## Abstract

Oxygen is an important and often limiting reagent of a firefly's bioluminescent chemical reaction. Therefore, the development of the tracheal system and its subsequent modification to support the function of firefly light organs are key to understanding this process. We employ micro-CT scanning, 3D rendering, and confocal microscopy to assess the abdominal tracheal system in *Photinus pyralis* from the external spiracles to the light organ's internal tracheal brush, a feature named here for the first time. The abdominal spiracles in firefly larvae and pupae are of the biforous type, with a filter apparatus and appear to have an occlusor muscle to restrict airflow. The first abdominal spiracle in the adult firefly is enlarged and bears an occlusor muscle, and abdominal spiracles two through eight are small, with a small atrium and bilobed closing apparatus. Internal tracheal system features, including various branches, trunks, and viscerals, were homologized across life stages. In adults, the sexually dimorphic elaboration and increase in volume associated with tracheal features of luminous segments emphasizes the importance of gas exchange during the bioluminescent process.

## Introduction

The light organs that produce bioluminescence in fireflies represent interesting evolutionary novelties and are as structurally diverse as the type of signals they produce. Fireflies (Coleoptera: Lampyridae) are a charismatic group of beetles that are commonly known for their ability to produce species-specific bioluminescent signals as adults. In Lampyridae, the type of luminous signals that are generated, the morphology of the light organ, and the ability to produce bioluminescence across life stages exhibit variability across species. This variation in both bioluminescent emissions and the light organs that produce them provides a rich source of data for the study of their evolution.

from Morphosource (https://www.morphosource.org/projects/000434376?locale=en).

**Funding:** Funding was provided by the National Science Foundation (https://www.nsf.gov/) M.A.B. DEB-1655936 in collaboration with K.F.S. DEB-1655908 and S.M.B. DEB-1655981 Funding for the AMNH Micro-CT scanner was provided via NSF instrumentation grant EAR-0959384 (https://www.nsf.gov/) H.W.H. was supported by a Richard Gilder Graduate School Fellowship (https://www.amnh.org/) The funders had no role in study design, data collection and analysis, decision to publish, or preparation of the manuscript.

**Competing interests:** The authors have declared that no competing interests exist.

To gain insight into the modifications of tissues that comprise these bioluminescent organs, we examine changes in the tracheal system across life stages (larva, pupa, and adults) of a single species, the North American firefly *Photinus pyralis* [1]. All known larvae are luminous and appear only capable of glowing, while adults show a range from nonluminous to either glowing or flashing [2–5]. *P. pyralis* possesses a simple glowing light organ in the larval stage that is not considered homologous with the more structurally complicated flashing light organs found in the adult males and females. By tracking tracheal system modifications across this beetle's metamorphosis, specifically in abdominal segments that initially lack and then gain light organs, we hope to gain clues into the abdominal modifications required to develop and support these amazing structures, and bioluminescent signals they produce.

## Appearance of the light organ

The appearance of the *P. pyralis* light organs differ by sex and life stage. In larvae, the light organs appear as a pair of small, spherical spots on abdominal segment VIII. These larval luminous emissions function as an aposematic signal to communicate their unpalatability to potential predators [6, 7]. The larval light organs produce a uniform glow that often rises and falls in intensity. In the early pupal stage, bioluminescence appears to originate from the larval light organs, which can remain functional up to a few days following eclosion into the adult stage. In the beginning of the pupal stage this light is emitted in the form of a glow, similar to what is seen in larvae. Further into pupation, the adult light organ may be seen functioning along with the larval light organ [8, 9]. Although it is unknown at how late in the pupal stage this occurs in *P. pyralis*, previous work on *Photuris pensylvanica* [10] indicates that both larval and adult light organs are functional in the same individual as early as 4–6 days after the last larval molt [9].

In adult male *P. pyralis*, the light organs cover the entire ventral surface of segments VI and VII (ventrites 5 and 6). These large light organs rest against the cuticle that is transparent to allow for the emission of light. In adult females the light organ appears as a somewhat spherical spot on the ventral surface of only segment VI. Adult male *P. pyralis* fireflies are commonly found flashing over open fields at dusk. While in flight, males produce a species-specific flash pattern of a single flash approximately every 6 seconds while flying in an upward "J" pattern. Females observe these flashing displays from perches on low vegetation, usually short grass, and respond to preferred males with a single flash that is specifically "aimed" at the selected male by turning the abdomen [11–13]. Within this flash-answer communication system male signals are presumed to be under selection to be visible at extended distances, while the female's response flash is directed towards select males [13, 14]. Aside from its use in courtship signaling, the bioluminescence of adults also serves an aposematic function [11, 12, 15–17].

## Tracheal systems

Insects have a respiratory system consisting of branching networks of tracheae that are connected to the outside of the body via spiracles. The branching of tracheae can differ by insect, species, and life stage. Elaborate and extensive tracheal branching networks are observed in the tracheal system of adult fireflies in flashing species while less extensive networks which taper into smaller tracheae occur in larvae, as well as adults that are nonluminous or only produce glows [8, 18–21]. Firefly tracheal systems are made up of a network of components, ranging from large to small, that include major tracheal trunks, tracheal branches, tracheal twigs, tracheoles, and tracheal end cells. Each of these components provides a pathway for respiratory gases to enter and exit the body, supplying the cells and tissues along the way. Similar to the arborization of tracheal branches that serve the flight muscles in pterygote insects [22], the

light organs of fireflies too are metabolically demanding in their need to move increased volumes of respiratory gases through the tracheae. In both cases, increased tracheation serves not only to deliver sufficient oxygen to meet the metabolic demands of the tissues, they also are needed to expel gaseous metabolic products so they do not hinder aerobic respiration. In comparison to glowing larvae, the light organs in flashing adult fireflies also possess a more elaborate/extensive tracheole system to meet the oxygen demands of producing bioluminescent flashes. While this expanded tracheation has been documented to include the development of a tracheal brush composed of many small tracheae which transport oxygen into the adult light organ tissues [18, 20, 21], a detailed analysis involving the comparison of homologized components across life stages of a single species has yet to be conducted. The modifications that occur in the firefly tracheal system across metamorphic life stages from larva to adult provides an exciting opportunity to study the modification, elaboration, and evolution of physiological features that contribute to the structure and function of firefly light organs.

### Purpose of study

The firefly's tracheal system, which is the primary respiratory system in insects, has become expanded to supply the flashing light organ of adult *P. pyralis* fireflies. Prior to this investigation, studies of the firefly tracheal system have been mostly limited by 2D imaging techniques and descriptions have remained very general with terminology such as the tracheal trunks, branches, and twigs [18, 19, 21, 23–25] and lack the required detail necessary to identify and track variation and modifications of specific features of this organ system. Additionally, few attempts have been made to quantify the tracheal system in fireflies due to the small size of the tracheae and the difficulty of measuring such a system in a 2D plane [18, 25]. In this study we utilized 3D data to compare features of the tracheal system across life stages of the *P. pyralis* firefly as they appear in situ. This method allows both accurate and detailed comparisons to be made across the larva, pupa, and adult life stages. The firefly light organ is a novel characteristic shaped by different evolutionary pressures and represents a unique and valuable opportunity to study character evolution in a single family. This work will not only further our understanding of firefly light organ morphology but provide a means of studying how tracheal morphology plays a role in developing and supporting a complex structure like the light organ.

## Materials and methods

### Specimen preparation

Individuals representing three life stages and both sexes (larva [final instar], male pupa [3–4 days old], adult male, and adult female) of *P. pyralis* were collected in Gladys, Virginia and New York, NY, USA. This species was selected due to its abundance, ease of collection, and extensive literature history. Live beetles were frozen in a -80˚ Celsius freezer in 1.5ml microtubes, which euthanized the beetles while preserving the tracheal system as it appears in situ. Specimens were thawed 30 minutes to 1 hour prior to scanning. Each specimen was placed inside a micropipette tube that was sealed to prevent movement and desiccation during the scan.

### Micro-CT scanning

The larva and pupa were scanned using a Phoenix v|tome|x s240 (GEs Measurement & Control business, Boston, MA) using a 180 kV X-ray tube. The larva was scanned with the following settings: a diamond target, 60 kV, 240 µA current, 400 ms detector time, a five-image average, and a 3.9 µm voxel resolution. The pupa was scanned with the following settings: a

diamond target, 70 kV, 285 µA current, 333 ms detector time, a four-image average, and a 7.3 µm voxel resolution.

The adult male and female were scanned using a Phoenix v|tome|x M using a 180 kV X-ray tube. The adult male was scanned with the following settings: a diamond target, 70 kV, 120 µA current, 500 ms detector time, a five-image average, and a 4.4 µm voxel resolution. The adult female was scanned with the following settings: a molybdenum target, 40 kV, 230 µA current, 500 ms detector time, a one-image average, and a 3.4 µm voxel resolution.

### 3D reconstruction and segmentation

Raw X-ray data for all four scans were reconstructed using GE's proprietary datos|x software v.2.3.2 and imported into Volume Graphics VG Studio Max v.3.0 to 3.5 (Volume Graphics, Heidelberg, Germany). Regions of interest (different segmentation regions) were created for abdominal segments V–VIII and for the segmented tracheal system. A wall thickness analysis was run on the segmented tracheal system to produce a color scheme that reflects tracheal lumen diameter. Each abdominal segment was separated into two regions of interest, one for the whole segment and the other for the tracheal system of that segment, which allowed for comparisons between segments in the same individual.

### Imaging and tracheal labelling

Images were taken of dorsal, lateral, and ventral views of each specimen's tracheal system within VG Studio Max. Additionally, the smaller tracheae were digitally removed from the adult male firefly to allow visualization of the tracheal system behind the extensive tracheal brush (all tracheae were included in volumetric measurements). Tracheal homologies were hypothesized based on prior studies and comparisons of tracheal systems in other insects (Table 1; Herhold et al. in prep.) [22, 26–28].

### Volumetric measurements

Regions of interest were created in VG Studio Max for each of abdominal segments V through VIII. This was also done for tracheal segmentation per segment. Volume measurements were taken for the total volume of each abdominal segment (including the space occupied by the tracheal system) and the volume of the tracheal system by itself (i.e., the empty tracheal lumen). The volume the tracheal system occupied within the total volume of the segment was calculated as a percentage. Percentages were used to allow for comparisons to be made

**Table 1. Tracheal branching abbreviations and terms.**

| Abbreviation | Name | Source |
|---|---|---|
| AS | Abdominal Spiracle | Raś et al. 2018 |
| SB | Spiracular Branch | New |
| DLT | Dorsal Longitudinal Trunk | Raś et al. 2018 |
| VLT | Ventral Longitudinal Trunk | Raś et al. 2018 |
| VC | Ventral Commissure | Raś et al. 2018; Snodgrass 1935 |
| Vi | Visceral | Šulc 1927; Barnhart 1961; Raś et al. 2018 |
| DVi | Dorsal Visceral | Raś et al. 2018 |
| VVi | Ventral Visceral | Raś et al. 2018 |
| Ga | Ganglion | Barnhart 1961; Snodgrass 1935 |
| x-Vi | Visceral branching from 'x' | New |

regardless of size variation between individuals.

$$(\text{Tracheal Volume}/\text{Segment Volume}) \times 100 = \text{Tracheal Volume Percent}$$

### Spiracle imaging

Using light and confocal microscopy, we imaged the spiracles on segments III and VI in the larva, pupa, adult male, and adult female, and segment I in the adult male of *P. pyralis* to visualize the structure of the spiracular opening and assess whether differences were present across life stages. The specimens were cleared using lactic acid and dissected. Images were taken using a Zeiss LSM 710 and collecting blue, green, and red spectra from autofluorescence of the cuticle.

## Results

### Tracheal descriptions by life stage

*Last Instar Larva* (Figs 1–3; Table 2). Abdominal spiracles of the biforous type and fairly uniform (Fig 4A and 4B). Each spiracle with elongate bicameral opening, the lateral margins of each opening with parallel branched trabeculae (Fig 4C and 4D); dendriform, multi-branched trabecular filter apparatus present before opening into bulbous atrium (Fig 4E and 4F); occlusor muscle extending from base of atrium (Fig 4F).

Spiracles with SB long in segments V–VIII, bearing three major tracheae—DLT, VVi, and VLT, which are similar in size. VLT connection incomplete between segments. VC and Ga long and thin, Ga much longer in segment VII than other segments. VC originates from base of VLT, near SB and branching with VVi. Major and minor branches with uniform diameter in segments V–VIII.

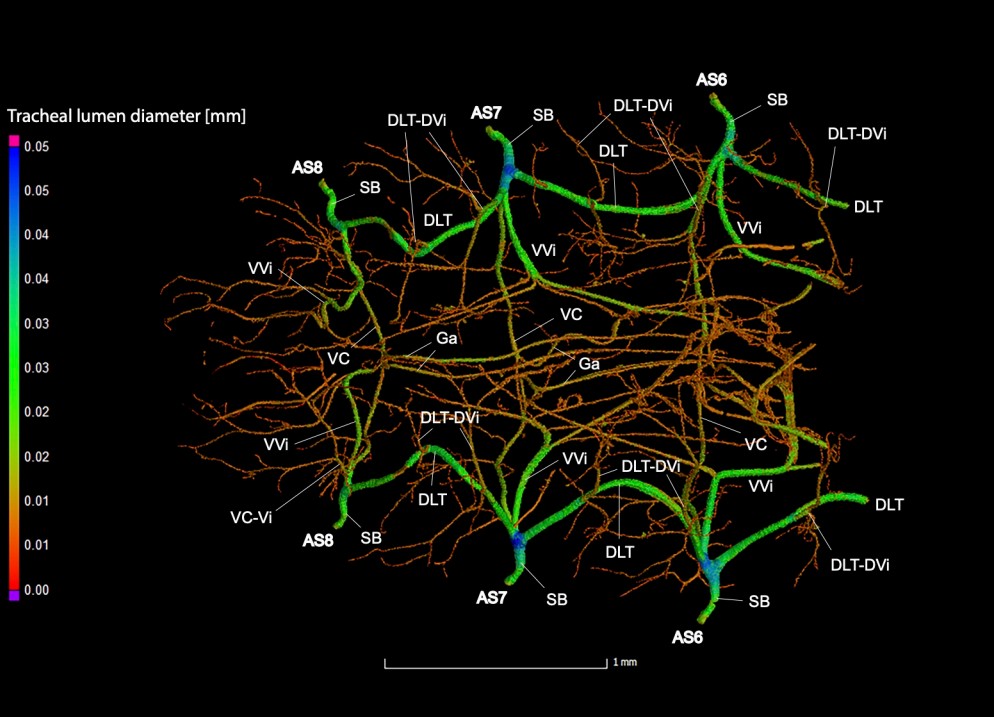

**Fig 1. Dorsal view of the abdominal tracheal system in *Photinus pyralis* final instar larva.**

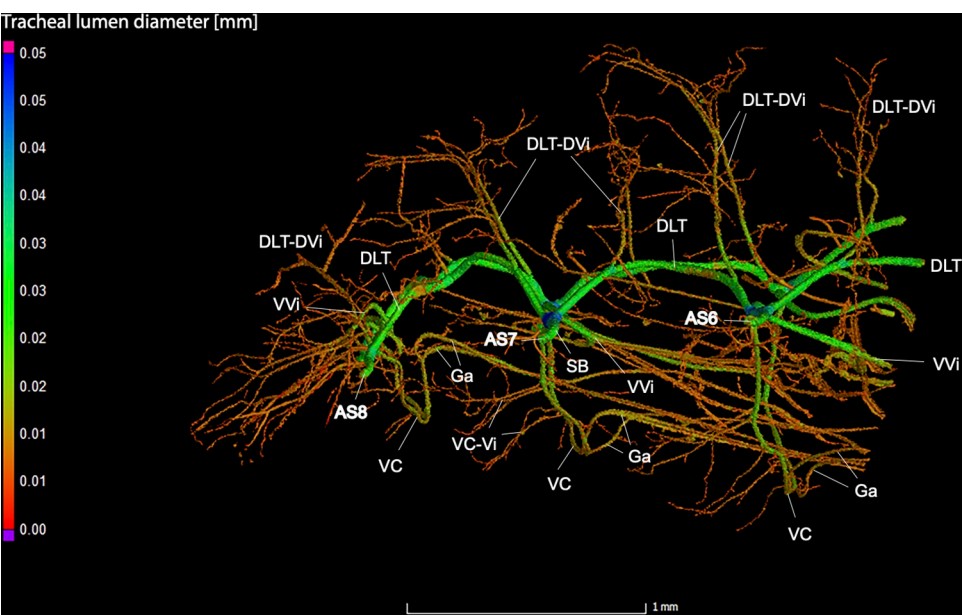

**Fig 2. Lateral view of the abdominal tracheal system in *Photinus pyralis* final instar larva.**

*3–4 Day Old Pupa* (Figs 5–7; Table 2). Abdominal spiracles similar to those of larva; biforous type (Fig 8A–8D); ecdysial scar and tube present.

Spiracles with SB long in segments V–VIII, with three major tracheae—DLT, VVi, and VLT. VLT connection incomplete between segments. VC uniform in size in segments V–VIII, Ga thin in all segments and longer in segment VII. VC originates from base of VLT at SB junction, slightly separated from base of VVi.

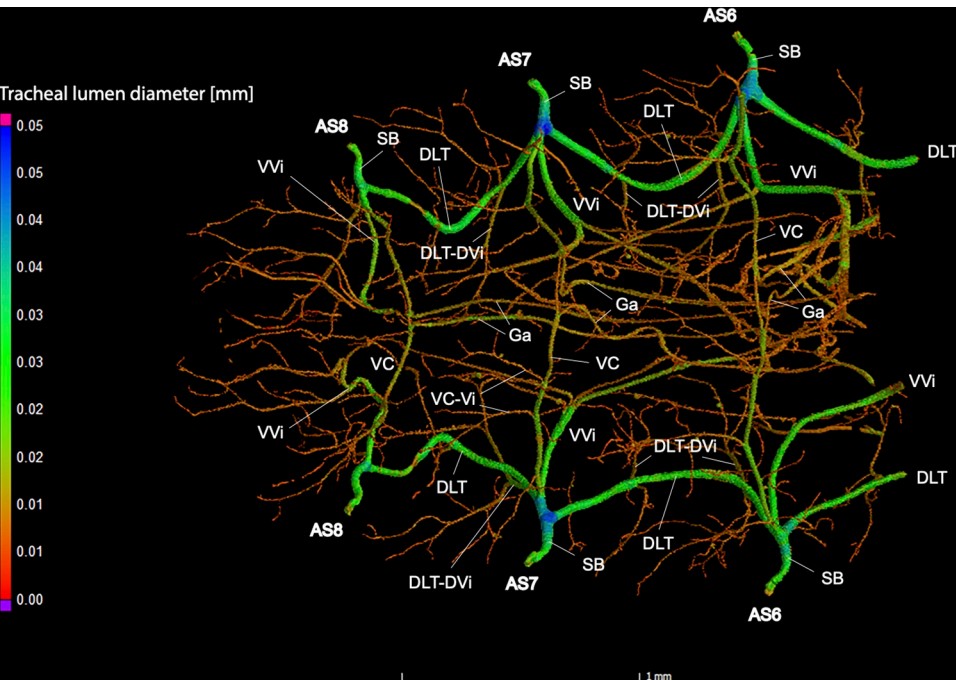

**Fig 3. Ventral view of the abdominal tracheal system in *Photinus pyralis* final instar larva.**

**Table 2. Comparisons of the major trunks, branches, and commissures of the abdominal tracheal system in *Photinus pyralis*.**

| Life Stage | Spiracular Branch (SB) | Dorsal Longitudinal Trunk (DLT) | Ventral Longitudinal Trunk (VLT) | Ventral Commissure (VC) | Ganglion Tracheae (Ga) |
|---|---|---|---|---|---|
| **Larva** | Long, thin, uniform in size in segments V–VIII | Uniform size, complete connection to SB | Incomplete connection between SB, uniform size | Thin, uniform size | Thin, uniform in size in segments V–VII, longest in segment VIII, originates from the VC |
| **Pupa** | Long, thin, uniform in size in segments V–VIII | Uniform size, complete connection to SB | Incomplete connection between SB, uniform size | Thin, uniform size | Thin, uniform size in segments V–VII, longest in segment VIII, originates from the VC |
| **Adult Male** | Short, larger in segments VI and VII | Uniform size, complete connection to SB | Complete connection between SB, larger in luminous segments VI and VII | Larger in luminous segments VI and VII | Thin in nonluminous segments, large in luminous segments, originates from the VC, longest in segment VIII |
| **Adult Female** | Short, uniform size in segments V–VIII | Uniform size, complete connection to SB | Complete connection between SB in some segments, larger in luminous segment VI | Larger in luminous segment VI | Thin, uniform size, originates from the VC, longest in segment VIII |

*Adult Male* (Figs 9–14; Table 2). Abdominal spiracles with AS1 large, distinct from remaining spiracles (Fig 15A). AS1 lacking large atrium and filter apparatus; occlusor muscle extending along spiracular margin (Fig 15B, 15D and 15E); lateral margin bearing occlusor muscle with medial joint possessing different cuticle, likely containing more resilin. Abdominal spiracles AS2–AS8 smaller, uniform, also lacking filter apparatus (Figs 15A and 16A–16D); a lobed pair of levers adjacent to tracheal opening internally; occlusor muscle extending between levers (similar in male and female; Fig 16F); cuticle of different composition between levers, likely with more resilin as in AS1 to facilitate flexion at this joint.

Spiracles with SB short, larger in luminous segments, bearing five major branches from three different tracheae—DLT, VLT, and VVi. DLT uniform in size in segments IV–VII. VLT larger in luminous segments VI and VII. VLT complete, extending from VC to SB. VVis medial to SB large in all segments, extending antero-medially; VVi external to SB extending posteriad, larger in luminous segments. VC and Ga larger in luminous segments. VLT-Vi and VC-Vi all larger in luminous segments. DLT-DVi present, similar size in all segments.

Comments. The male has complete dorsal and ventral longitudinal trunks connecting each abdominal spiracle. Most branches are enlarged in the luminous segments (sternites 6–7), including those branching from AS6–8, although the VC extending from AS8 is not enlarged (Table 4; Figs 27 and 28). The VVi extending internally from SB are approximately the same in all segments, as are the DLT-DVi.

*Adult Female* (Figs 17–19; Table 2). Abdominal spiracles not noticeably different from those of male (Fig 16E and 16F).

Spiracles with SB short, bearing five major branches from three tracheae—DLT, VLT, and VVi. DLT uniform diameter throughout segments V–VIII. DLT uniform in size in segments IV–VII. VLT larger in luminous segment VI. VLT complete, extending from VC to SB. VC and Ga larger in luminous segment VI, Ga also larger in nonluminous segment VII. VLT slightly larger in luminous segment. VLT-Vi and VC-Vi enlarged in luminous segment. DLT-DVi similar size in all segments.

## Individual variation

Three additional adult male *P. pyralis* were prepared and scanned to test for variation within the species. No notable variation was observed in the tracheal features specifically examined in this study. This provided confidence that those specimens included in these analyses were accurate representatives of their life stage and species despite a small sample size (unpublished data).

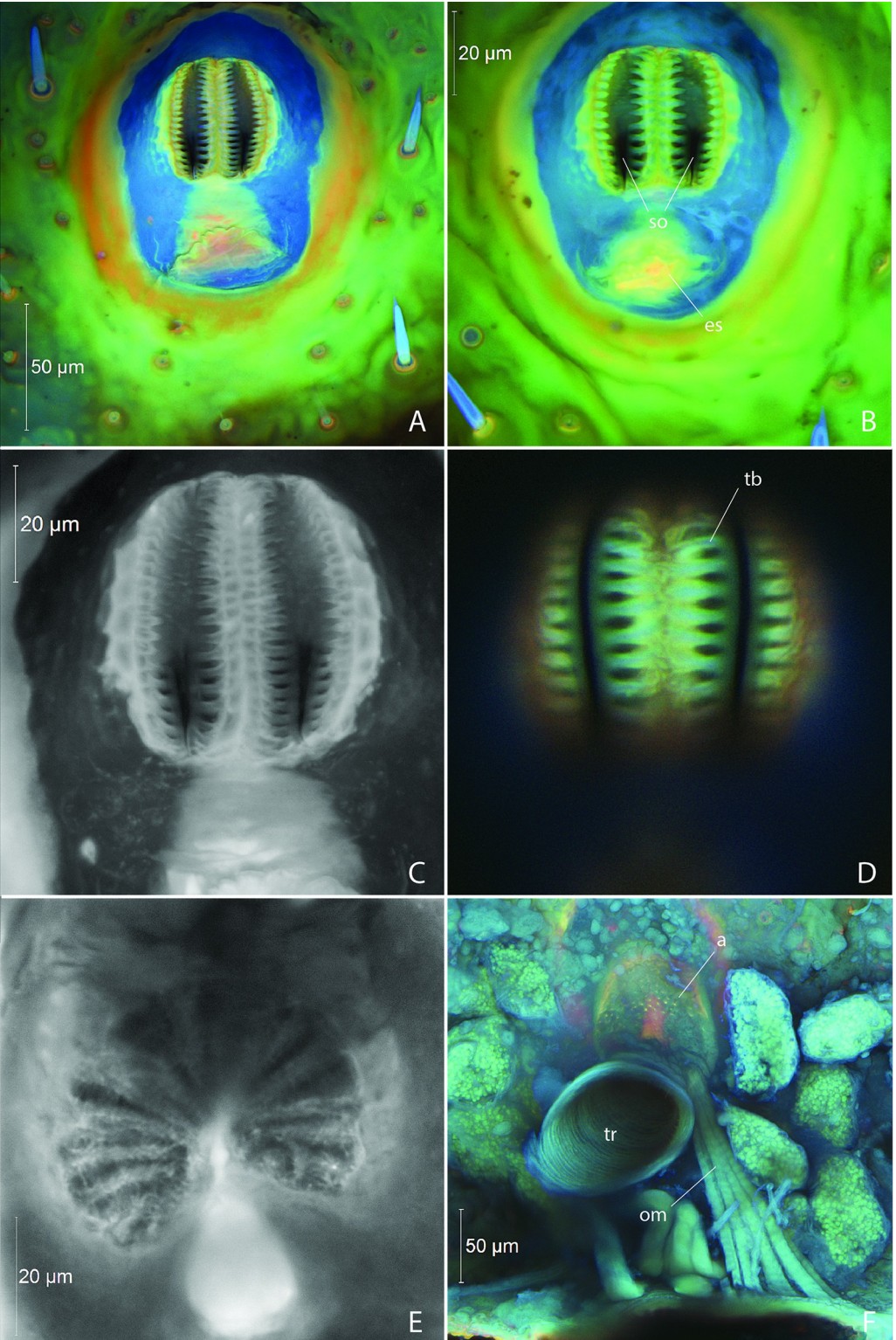

**Fig 4. Larval abdominal spiracles in *Photinus pyralis*.** A, segment III spiracle, external view; B, segment VI spiracle, external view; C, segment VI spiracle, enlargement of external view; D, segment VI spiracle, enlargement showing parallel trabeculae along lateral margins of spiracular openings; E, segment VI spiracle, optical section between spiracular opening and atrium, showing dendriform filter apparatus; F, segment VI spiracle, internal view showing atrium and occlusor muscle. Abbreviations: a, atrium; es, ecdysial scar; om, occlusor muscle; so, spiracular opening; tb, trabeculae; tr, trachea.

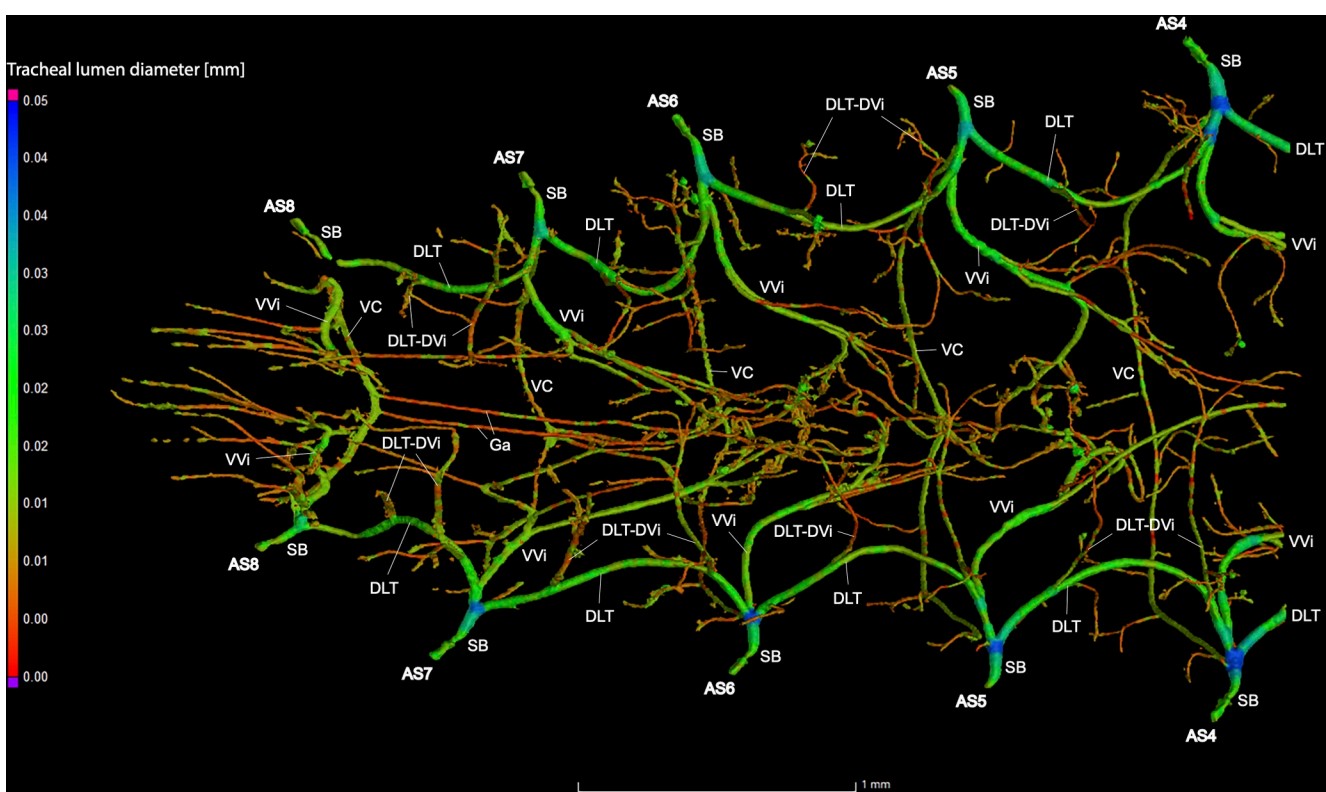

**Fig 5. Dorsal view of the abdominal tracheal system in *Photinus pyralis* 3–4 day-old pupa.**

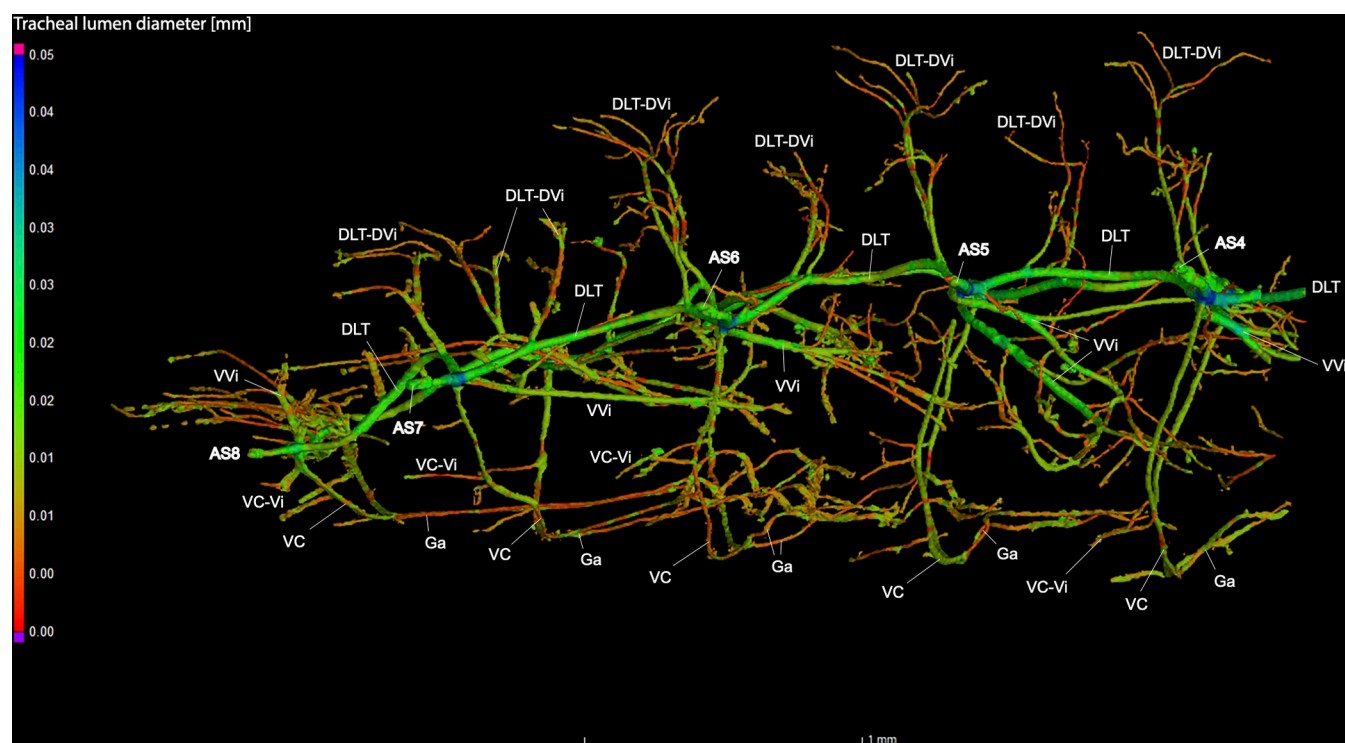

**Fig 6. Lateral view of the abdominal tracheal system in *Photinus pyralis* 3–4 day-old pupa.**

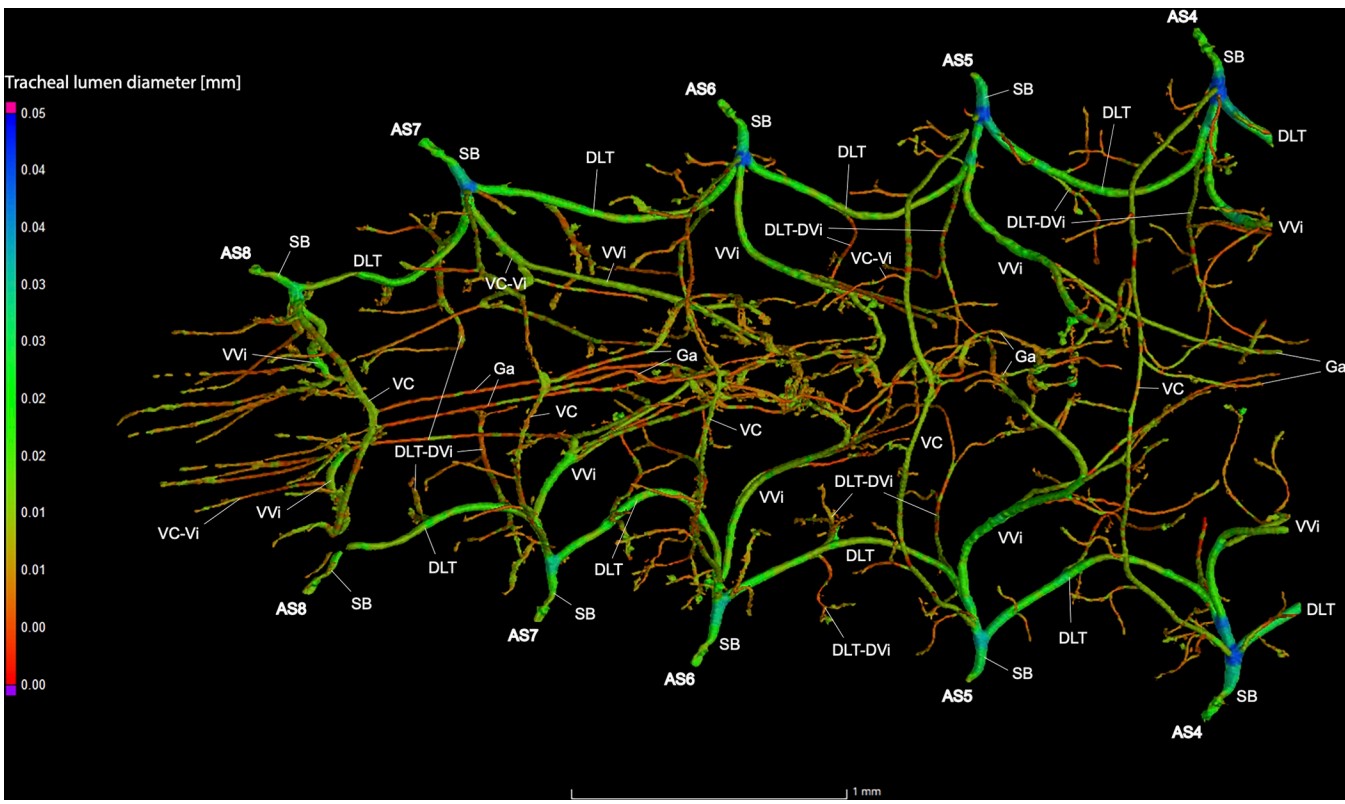

**Fig 7. Ventral view of the abdominal tracheal system in *Photinus pyralis* 3–4 day-old pupa.**

## Volumetric measurements of the tracheal system

Abdominal tracheation was measured by volume in segments V–VIII in the larva, pupa, adult male, and adult female (Table 3, Figs 20 and 21). Results indicate that there is similar tracheal volume in the last instar larva and 3–4 day-old pupa (0.29% in segments V–VIII and 0.35% for segments V–VIII, respectively) with only a 0.06% increase between life stages. When the pupa progresses to the adult female (3.01% tracheal volume for segments V–VIII) there is a 2.66% increase in the subsequent life stage. The greatest difference in tracheal volume (5.47% increase) is seen between the pupa and the adult male (5.82% tracheal volume for segments V–VIII).

## Major Differences of the tracheal system across life stages

**Spiracular structure.** In the larva, the abdominal spiracles are of the biforous type [29], possessing a narrow bicameral opening lined by parallel trabeculae (Fig 4A and 4B). Further into the spiracular opening is a dendriform filter apparatus, followed by a bulbous atrium lined with setae internally (Fig 4C–4E). The occlusor muscle inserts at one side of the junction between the atrium and trachea (Fig 4E). The ecdysial tube splits from the atrium and connects to the external body wall at the ecdysial scar. The pupal (3–4 days old) abdominal spiracles are similar to those of the larva, the only difference perhaps being a slightly smaller atrium and filter apparatus (Fig 8A–8D). The size of the abdominal spiracular openings do not appear to vary significantly between the larva and pupa.

The adult abdominal spiracles are of two types, with a large spiracle on abdominal segment I and small spiracles on segments II–VIII (Figs 15A–15E and 16A–16F). Both types have little

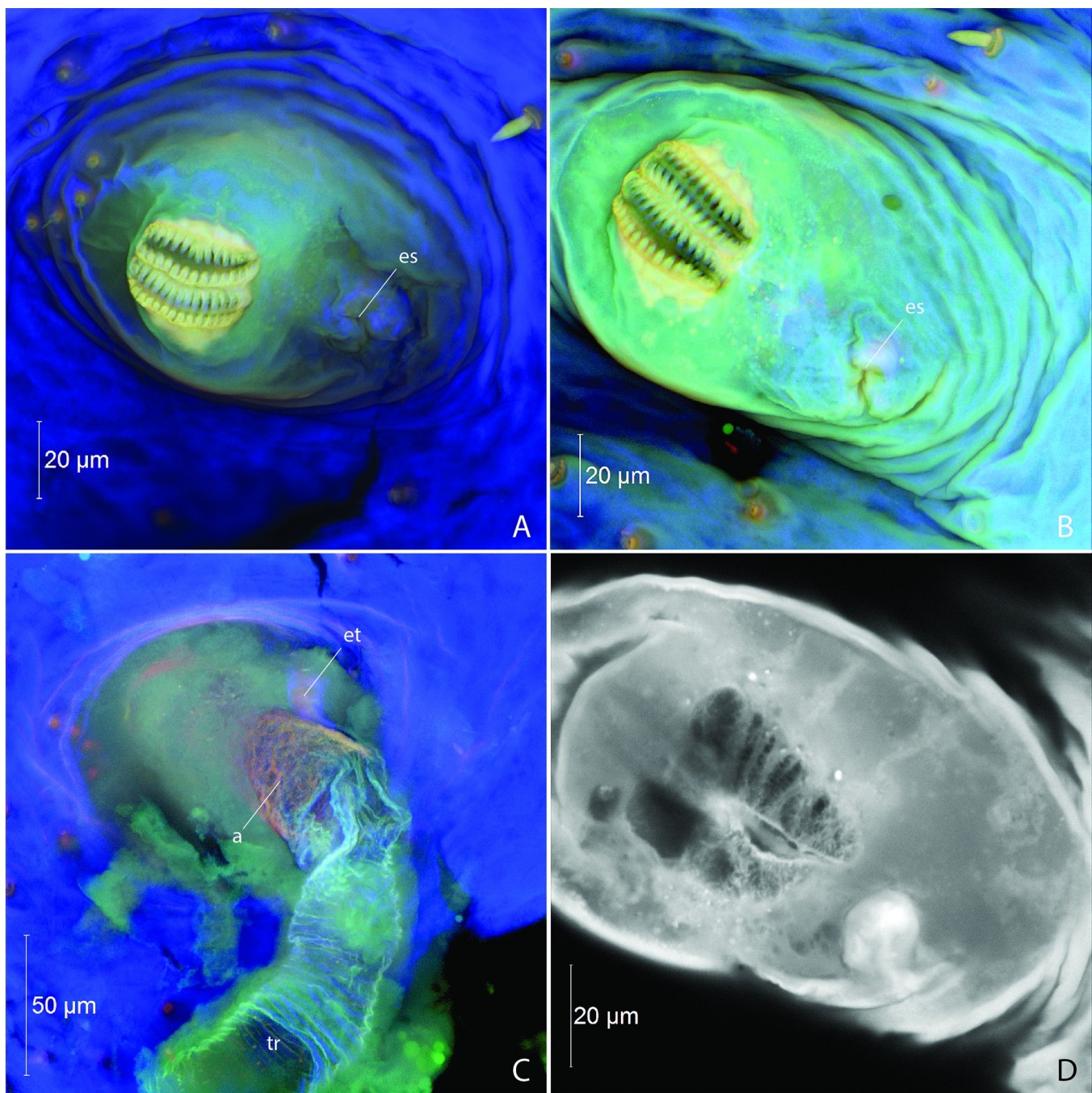

**Fig 8. Pupal abdominal spiracles in *Photinus pyralis*.** A, segment III spiracle, external view; B, segment VI spiracle, external view; C, segment VI spiracle, internal view; D, segment VI spiracle, optical section between spiracular opening and atrium, showing dendriform filter apparatus. Abbreviations: a, atrium; es, ecdysial scar; et, ecdysial tube; tr, trachea.

to no atrium. The larger first abdominal spiracle is typical in Coleoptera [29], its branches largely penetrating the thoracic flight muscles [22]. It is an elongated slit-like opening which directly connects to the SB trachea and it possesses a lateral occlusor muscle (Fig 15E). Along the margin of the spiracular opening bearing the occlusor muscle the cuticle appears to have a different composition at the region of flexion (Fig 15E, indicated by arrow). As resilin can be

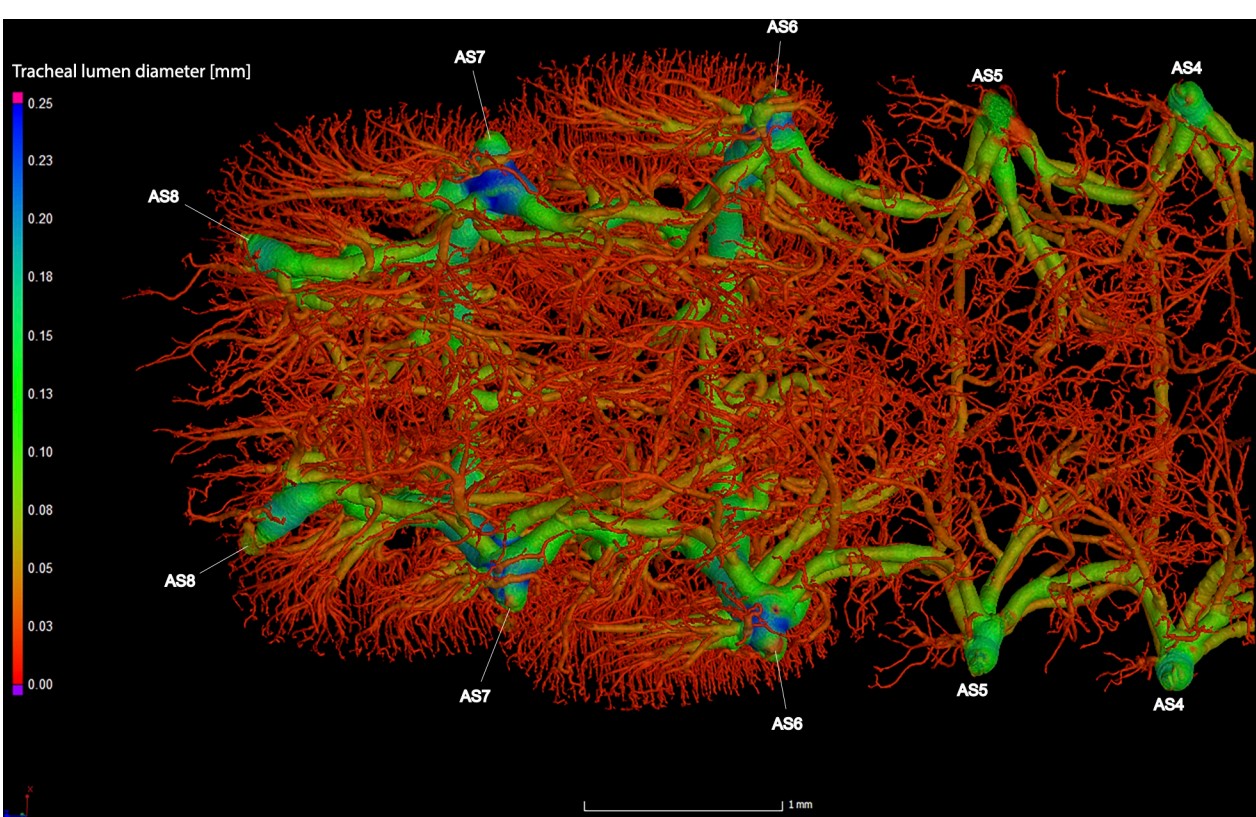

**Fig 9. Dorsal view of the abdominal tracheal system in adult male *Photinus pyralis*.**

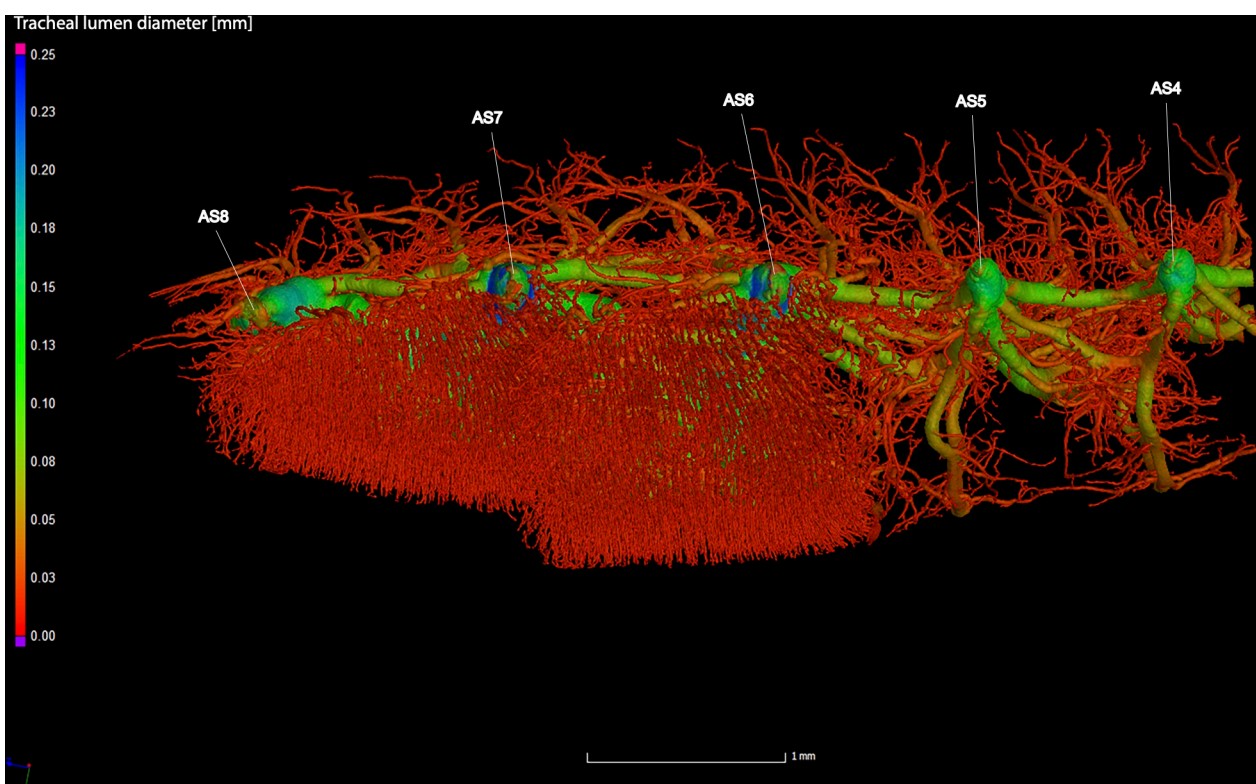

**Fig 10. Lateral view of the abdominal tracheal system in adult male *Photinus pyralis*.**

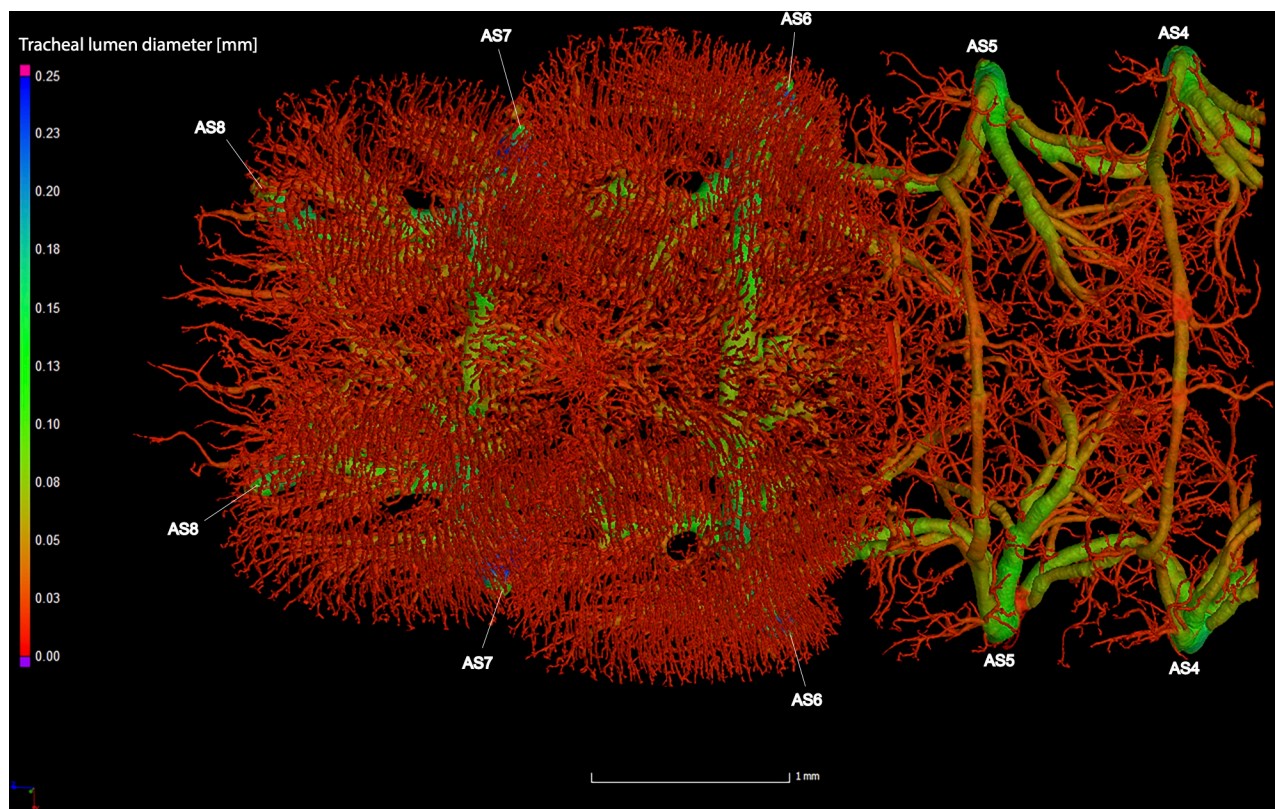

**Fig 11. Ventral view of the abdominal tracheal system in adult male *Photinus pyralis*.**

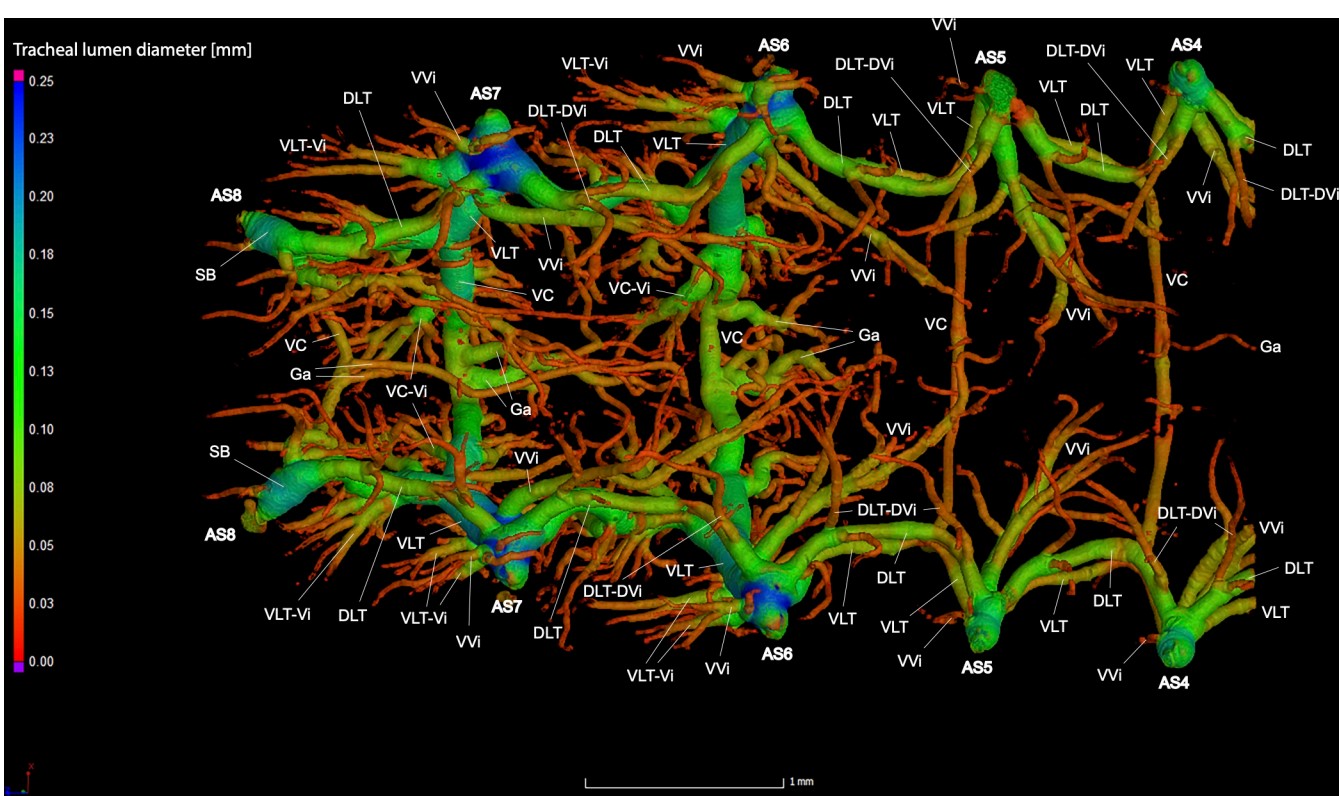

**Fig 12. Dorsal view of the abdominal tracheal system in adult male *Photinus pyralis* with smaller tracheae digitally removed.**

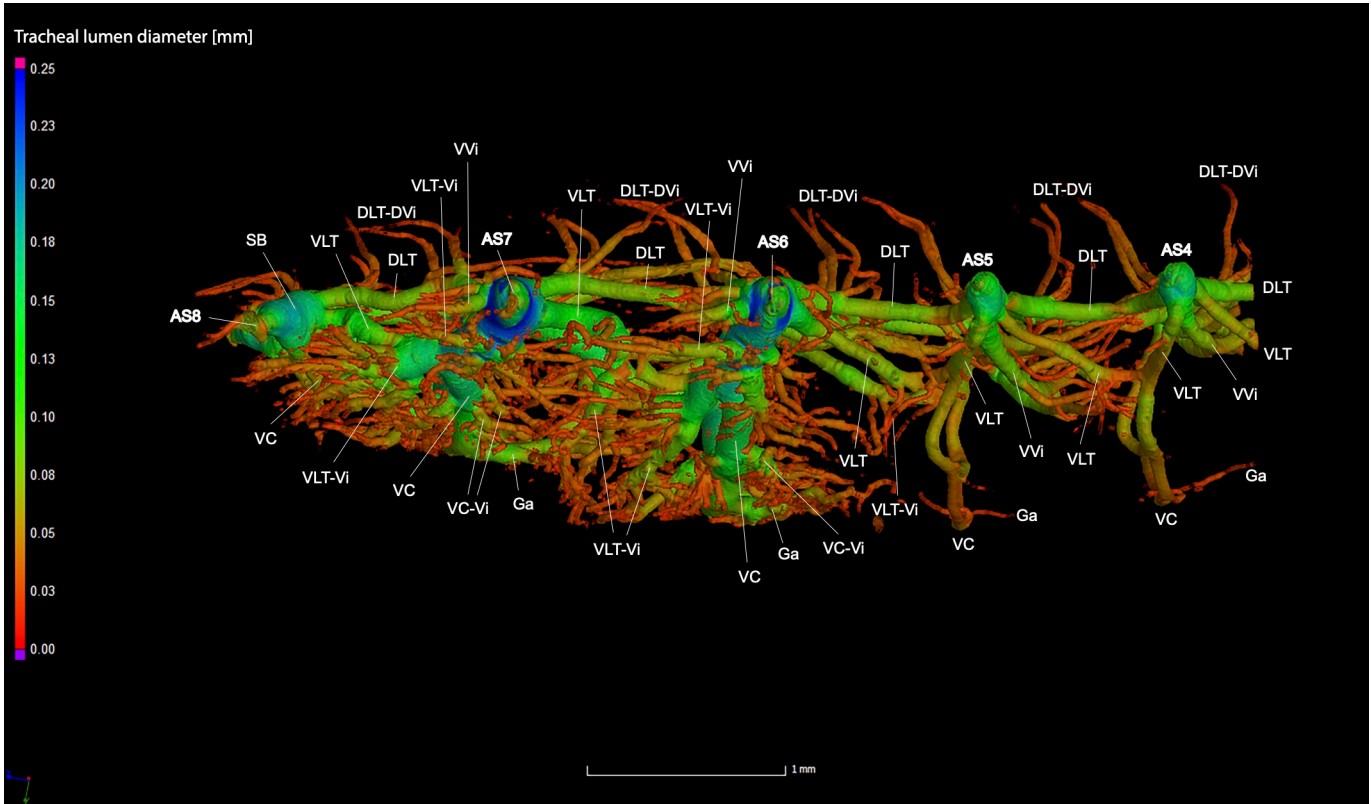

**Fig 13. Lateral view of the abdominal tracheal system in adult male *Photinus pyralis* with smaller tracheae digitally removed.**

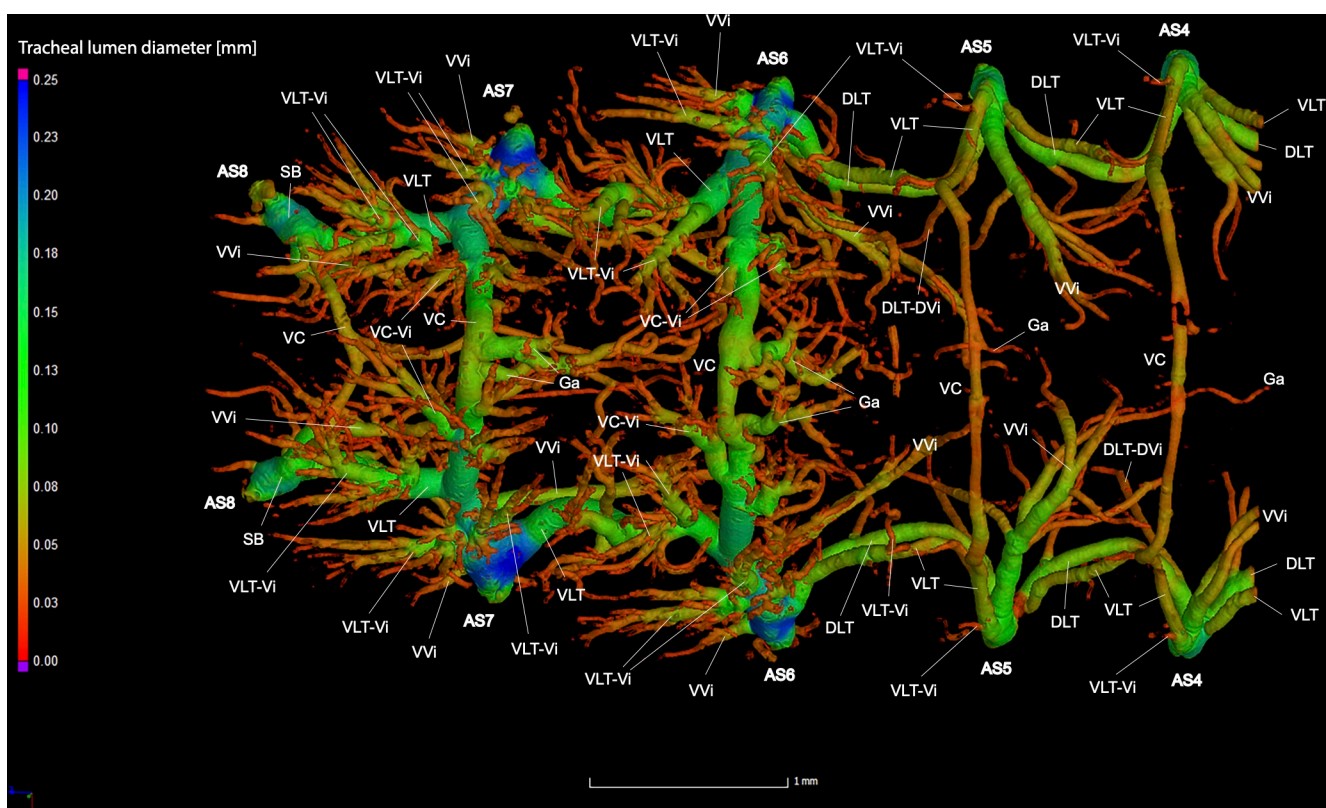

**Fig 14. Ventral view of the abdominal tracheal system in adult male *Photinus pyralis* with smaller tracheae digitally removed.**

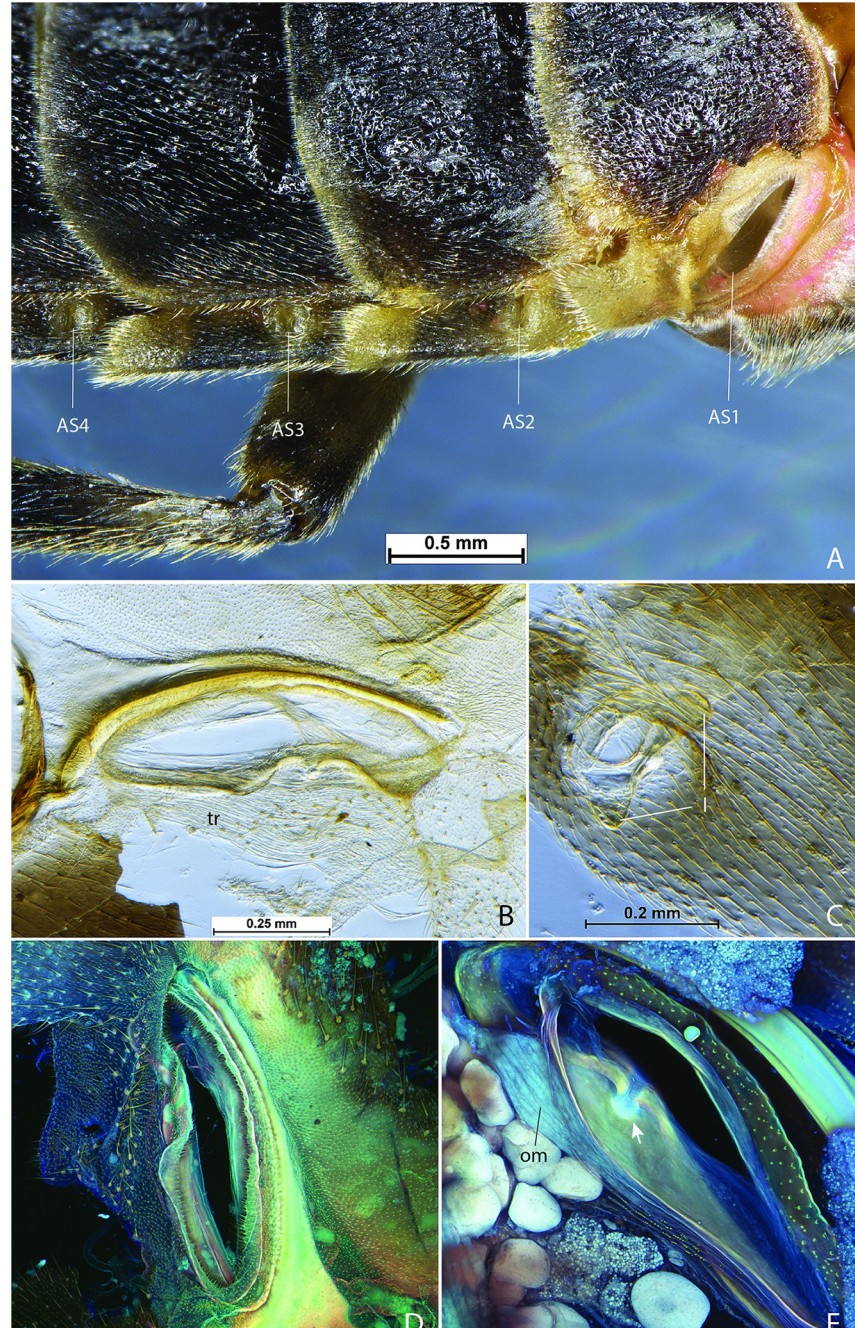

**Fig 15. Adult male abdominal spiracles in *Photinus pyralis*.** A, right half of abdomen, dorsal view showing spiracles 1–4; B, segment I spiracle, external view; C, segment III spiracle, external view; D, segment I spiracle, external view; E, segment I spiracle, internal view showing occlusor muscle. Abbreviations: l, lever; om, occlusor muscle; tr, trachea.

visualized by capturing a certain range of blue-spectrum autofluorescence [30], this area likely contains more resilin to allow flexion (closing) of the spiracular lip. In contrast to this large anterior spiracle, the smaller spiracles on segments II–VIII possess a short atrium that leads to an internal pair of digitiform lever arms (the closing mechanism) connected by the occlusor muscle [31] (Figs 16F, 22A and 22B). Between the lever arms is an area of different cuticular

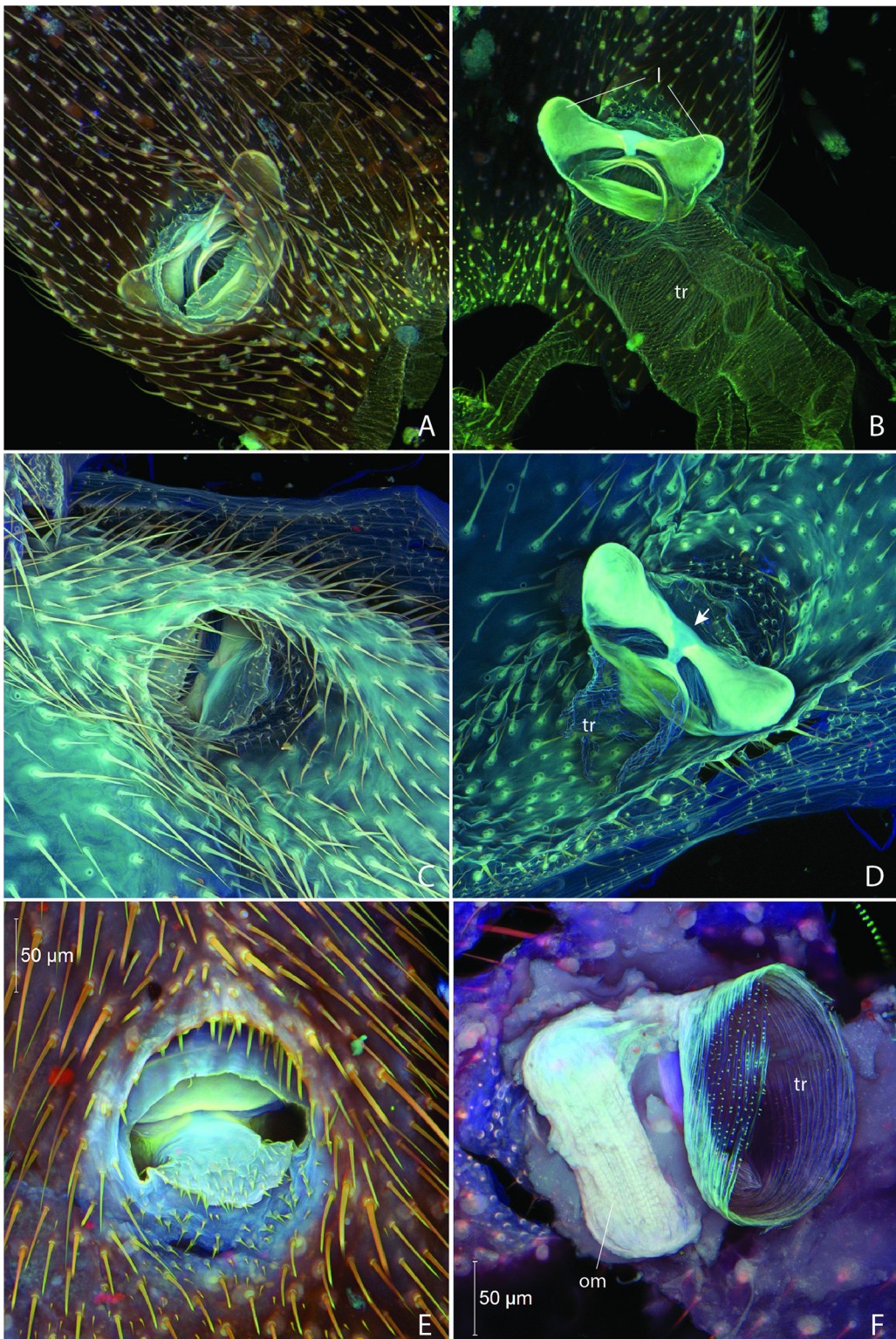

**Fig 16.** Adult male (A–D) and female (E–F) abdominal spiracles in *Photinus pyralis*. A, segment III spiracle, external view; B, segment III spiracle, internal view; C, segment VI spiracle, external view; D, segment VI spiracle, internal view; E, segment VI spiracle, external view; F, segment VI spiracle, internal view showing occlusor muscle. Abbreviations: l, lever; om, occlusor muscle; tr, trachea.

composition (Fig 16D, indicated by arrow). Similar to AS1, this area likely possesses more resilin to accommodate flexion (closing) at the joint. The spiracular atrium was visible in segment VIII of the adult male micro-CT scan, which provided the opportunity to visualize the levers of the closing mechanism and the structure between the spiracle and spiracular branch (SB) (Fig 22A and 22B). The closing mechanism features two hollow lobes that are connected to each other by muscle. These lobes are present just inside the spiracle at a point of constriction and function to manually open and close the spiracle. Beyond this constriction is the much larger SB. Although the abbreviated spiracular branch becomes enlarged in the adult male luminous segments, spiracular size does not appear to noticeably vary among segments II–VIII.

**Ventral and dorsal longitudinal trunks.**    The ventral longitudinal trunks (VLT) were analyzed in abdominal segments IV–VII in the adult male and female (Figs 23–26, Table 4). Measurements were taken on the right side of the body, with the exception of segments IV and VI in the female. This was due to part of the tracheae filling with fluid prior to scanning (segment IV) and a structurally incomplete connection of the VLT (segment VI) on the right side of the abdomen. The VLT connection is also incomplete in the left side of segment VII in the female. The volume of the VLT changed considerably between luminous and nonluminous segments in both male and female adults (Fig 27). The dorsal longitudinal trunks (DLT) were segmented as well to compare to the VLT. The volume of the DLT did not show a major increase in the luminous segments of either sex.

**Ventral commissures.**    The ventral commissures were segmented and analyzed in both male and female adults and results indicate an increase in volume of luminous segments (segment VI in female, VI and VII in male) (Figs 23–26 and 28; Table 4). The ventral commissures

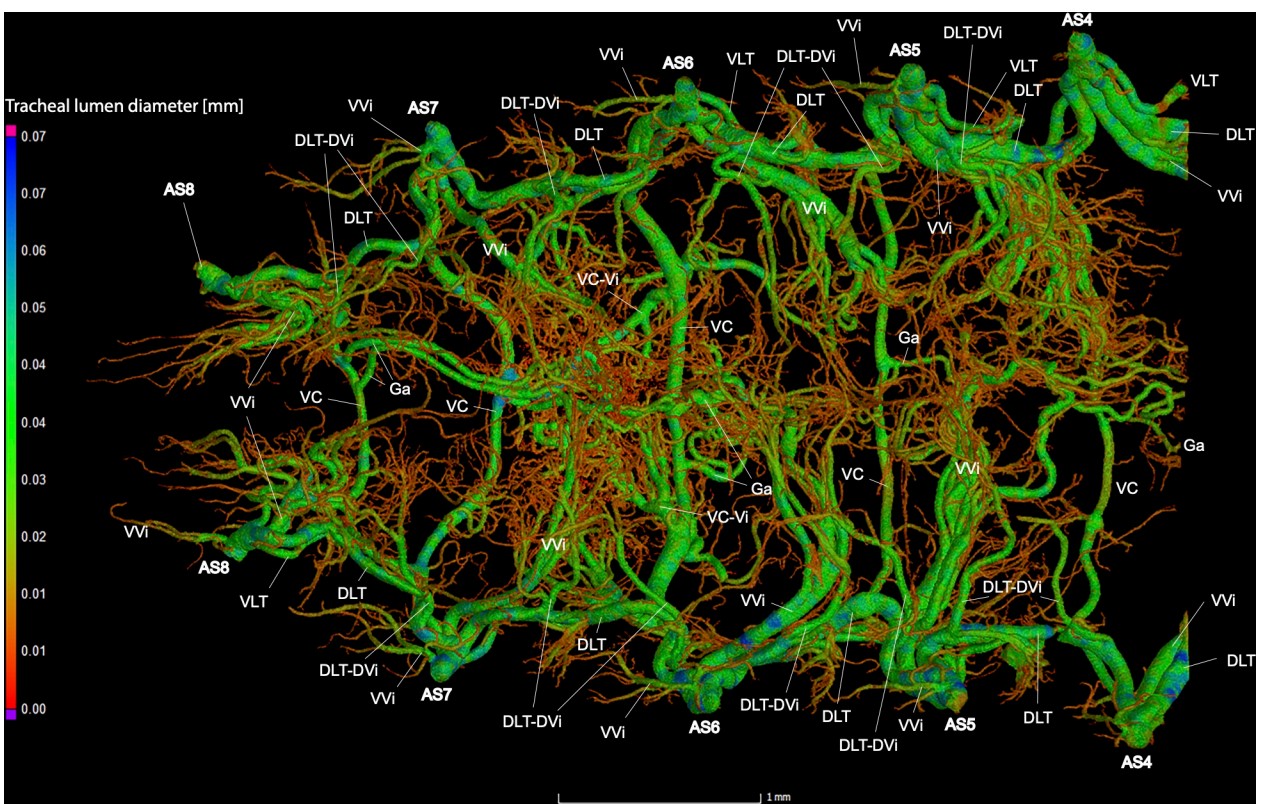

**Fig 17. Dorsal view of the abdominal tracheal system in adult female *Photinus pyralis*.**

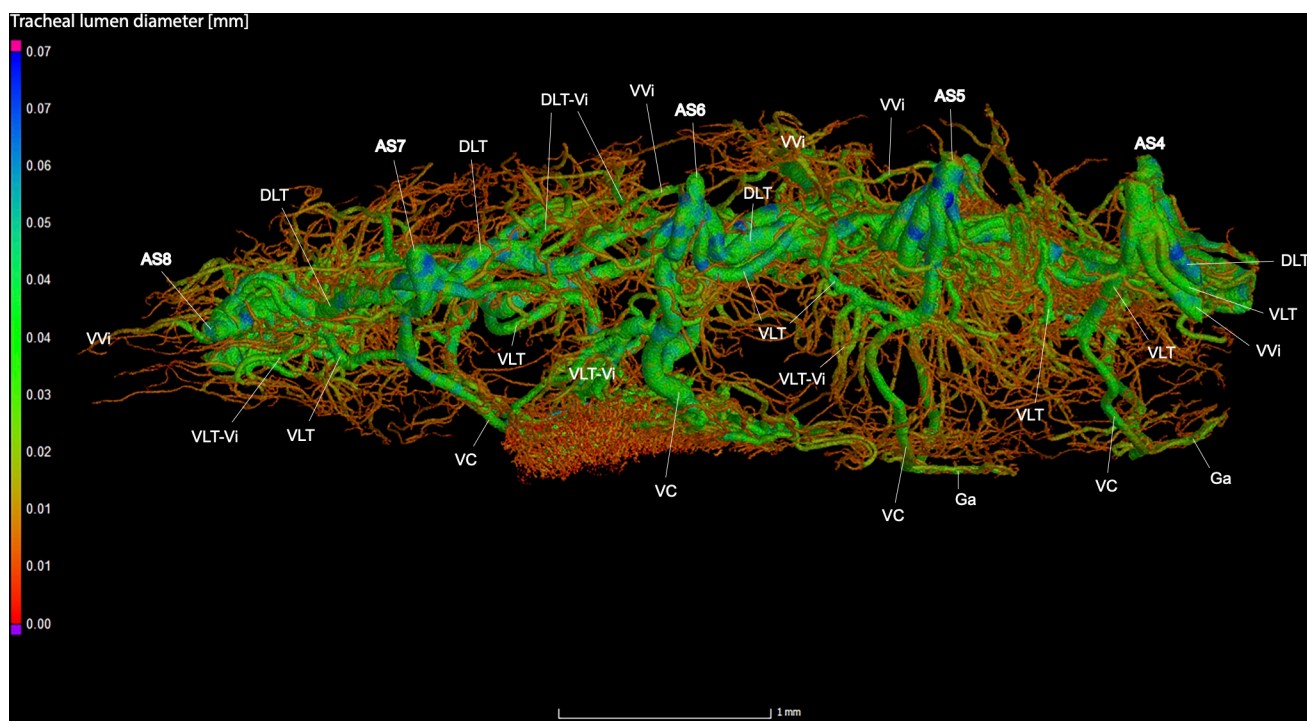

**Fig 18. Lateral view of the abdominal tracheal system in adult female *Photinus pyralis*.**

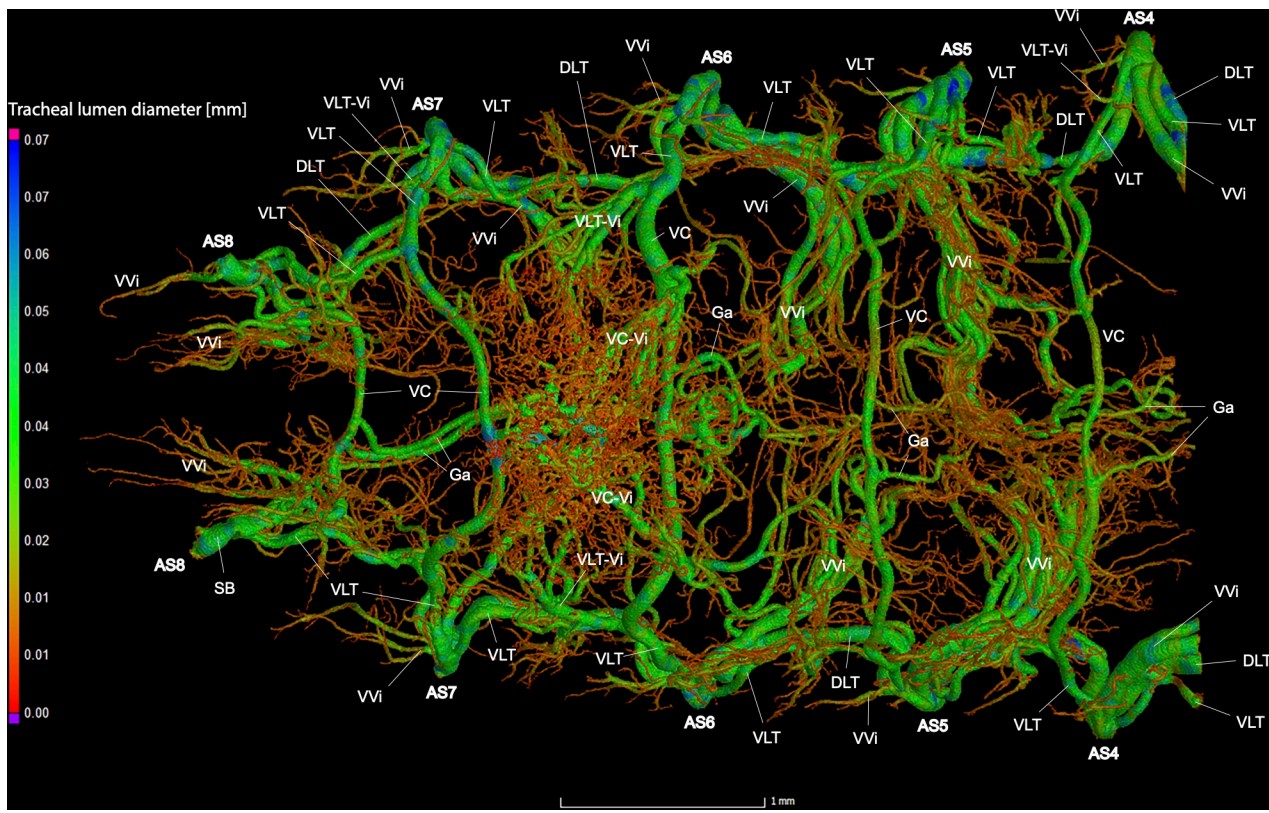

**Fig 19. Ventral view of the abdominal tracheal system in adult female *Photinus pyralis*.**

**Table 3. Volumetric measurements by segment in larva (last instar), pupa (3–4 days old), adult male, and adult female *Photinus pyralis*.**

| Life Stage | Segment | Total Segment Volume (mm³) | Segment Tracheal Volume (mm³) | Tracheal Volume Percent (%) |
|---|---|---|---|---|
| **Larva** | **V** | 4.8519 | 0.0133 | **0.2735** |
| | **VI** | 2.7078 | 0.0084 | **0.3084** |
| | **VII** | 0.6714 | 0.0013 | **0.1996** |
| | **VIII** | 0.6269 | 0.0027 | **0.4355** |
| | **Total V–VIII** | 8.8579 | 0.0257 | **0.2900** |
| **Pupa** | **V** | 2.9382 | 0.0089 | **0.3029** |
| | **VI** | 2.2999 | 0.0093 | **0.4057** |
| | **VII** | 1.5136 | 0.0058 | **0.3819** |
| | **VIII** | 1.0717 | 0.0035 | **0.3247** |
| | **Total V–VIII** | 7.8233 | 0.0275 | **0.3514** |
| **Adult Male** | **V** | 4.4402 | 0.1197 | **2.6965** |
| | **VI** | 6.8883 | 0.4498 | **6.5299** |
| | **VII** | 4.4267 | 0.3786 | **8.5536** |
| | **VIII** | 0.9472 | 0.0245 | **2.5834** |
| | **Total V–VIII** | 16.7023 | 0.9726 | **5.8234** |
| **Adult Female** | **V** | 4.0561 | 0.1345 | **3.3163** |
| | **VI** | 4.3154 | 0.1355 | **3.1390** |
| | **VII** | 2.5174 | 0.0705 | **2.8005** |
| | **VIII** | 1.2851 | 0.0257 | **1.9983** |
| | **Total V–VIII** | 12.174 | 0.3662 | **3.0076** |

Segments with a functional light organ are highlighted in yellow. Values in the Total columns include the sum of total segment volume in V–VIII, the sum of segment tracheal volume in V–VIII, and the percentage of tracheal volume of the totals for segments V–VIII.

give rise to much of the tracheal brush that supplies oxygen to the light organ, as well as other visceral tracheation (VC-Vi) and tracheae supplying the ganglia (Ga). Although the male has much greater increase in volume of the VC in luminous segments, there is an increase in the luminous segment of the female as well.

**Tracheal brush.** The adult male and female possess an elaborate tracheal brush, a high concentration of tracheae that supply oxygen to the light organ tissues (Figs 29, 30 and 31A–31D). The tracheal brush is a feature that has been noted in many species and appears correlated with the presence of a flashing light organ [21]. Its terminal tracheal trunks are nearly parallel to one another and run orthogonal to the light organ tissue, penetrating the photogenic layer. These terminal trunks comprise the center of each tracheal end organ cylinder,

**Table 4. Volume of Tracheal Features in *Photinus pyralis* (mm³).**

| | IV | V | VI (L.O. both sexes) | VII (L.O. male only) | VIII |
|---|---|---|---|---|---|
| **Adult Male DLT** | R: 0.0075 | R: 0.0074 | R: 0.0084 | R: 0.0056 | - |
| **Adult Female DLT** | R: 0.0097 | R: 0.0096 | R: 0.0082 | R: 0.0052 | - |
| **Adult Male VLT** | R: 0.0045 | R: 0.0071 | R: 0.0226 | R: 0.0188 | - |
| **Adult Female VLT** | L: 0.0046 | R: 0.0054 | L: 0.0072 | R: 0.0028 | - |
| **Adult Male VC** | 0.0082 | 0.0132 | 0.0329 | 0.0248 | 0.0038 |
| **Adult Female VC** | 0.0050 | 0.0051 | 0.0140 | 0.0060 | 0.0030 |

Volume of the dorsal longitudinal trunks (DLT; orange) and ventral longitudinal trunks (VLT; green) in segments IV–VII, and of the ventral commissures (VC; blue) in segments IV–VIII of adult male and female *Photinus pyralis*. L and R indicate if the measurement was taken on the left or right side of the body. The spiracular branches (SB) end on segment VIII, therefore there is no complete connection of DLT and VLT between SB beyond segment VII. "L.O." indicates the presence of a light organ.

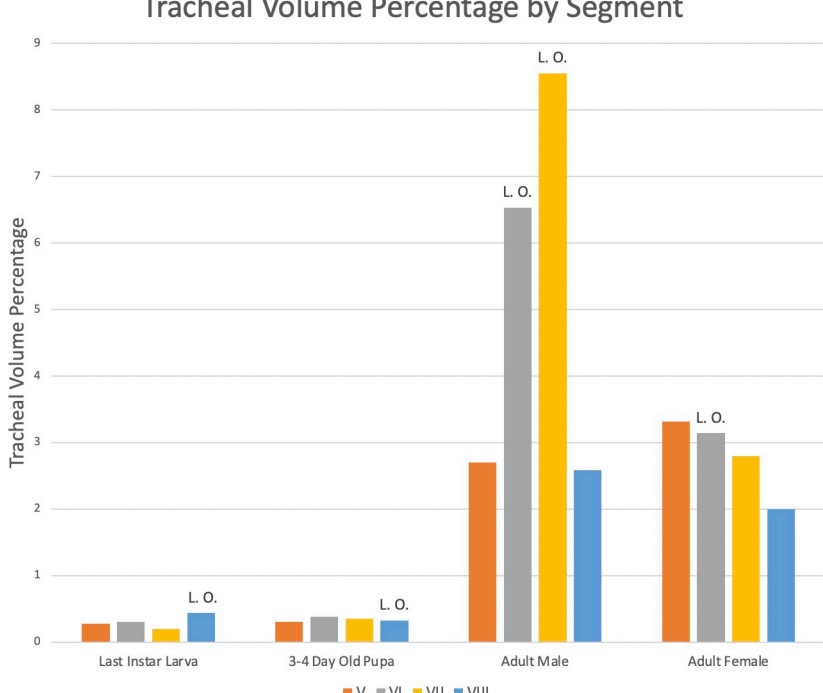

**Fig 20. Tracheal volume percent by abdominal segment in larva, pupa, adult male, and adult female *Photinus pyralis*.** "L. O." indicates the presence of a light organ on that segment.

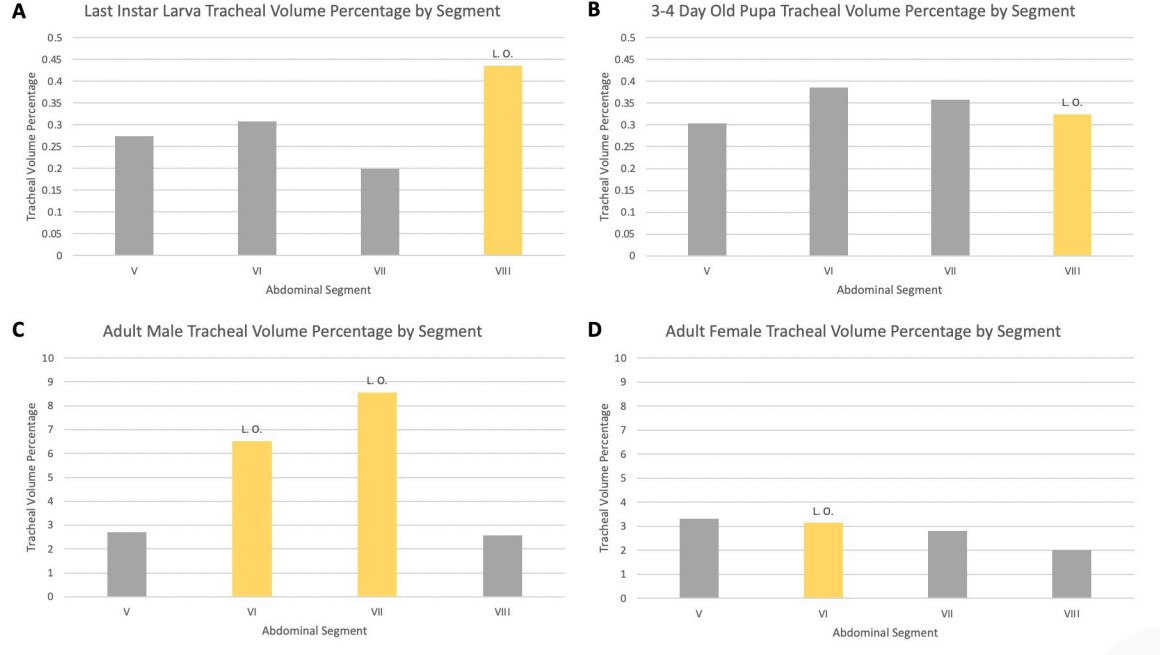

**Fig 21.** Tracheal volume percent by segment in larva (A), pupa (B), adult male (C), and adult female (D) *Photinus pyralis*. "L. O." indicates the presence of a functional light organ on that segment.

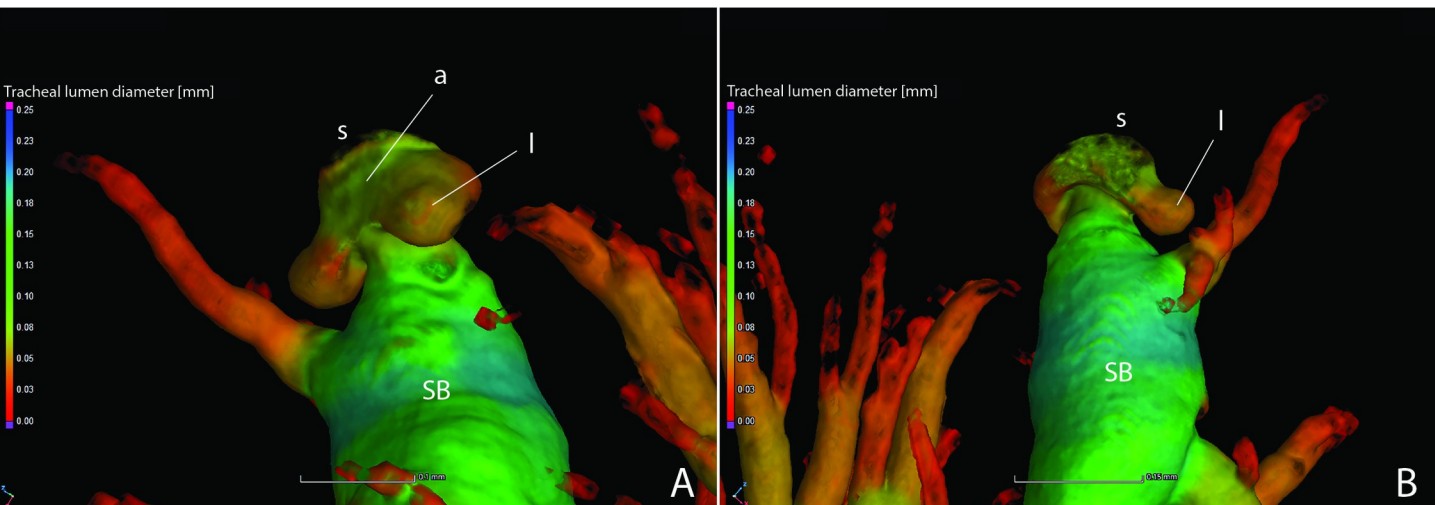

**Fig 22. The spiracle of abdominal segment VIII in adult male *Photinus pyralis*.** Internal view, ventrally (A) and dorsally (B). Abbreviations: a, atrium; l, lever; s, spiracle; SB, spiracular branch.

**Fig 23. Dorsal view of the adult male *Photinus pyralis* tracheal system.** Ventral commissures (VC; blue), dorsal longitudinal trunks (DLT; orange), and ventral longitudinal trunks (VLT; green) highlighted for volumetric analysis.

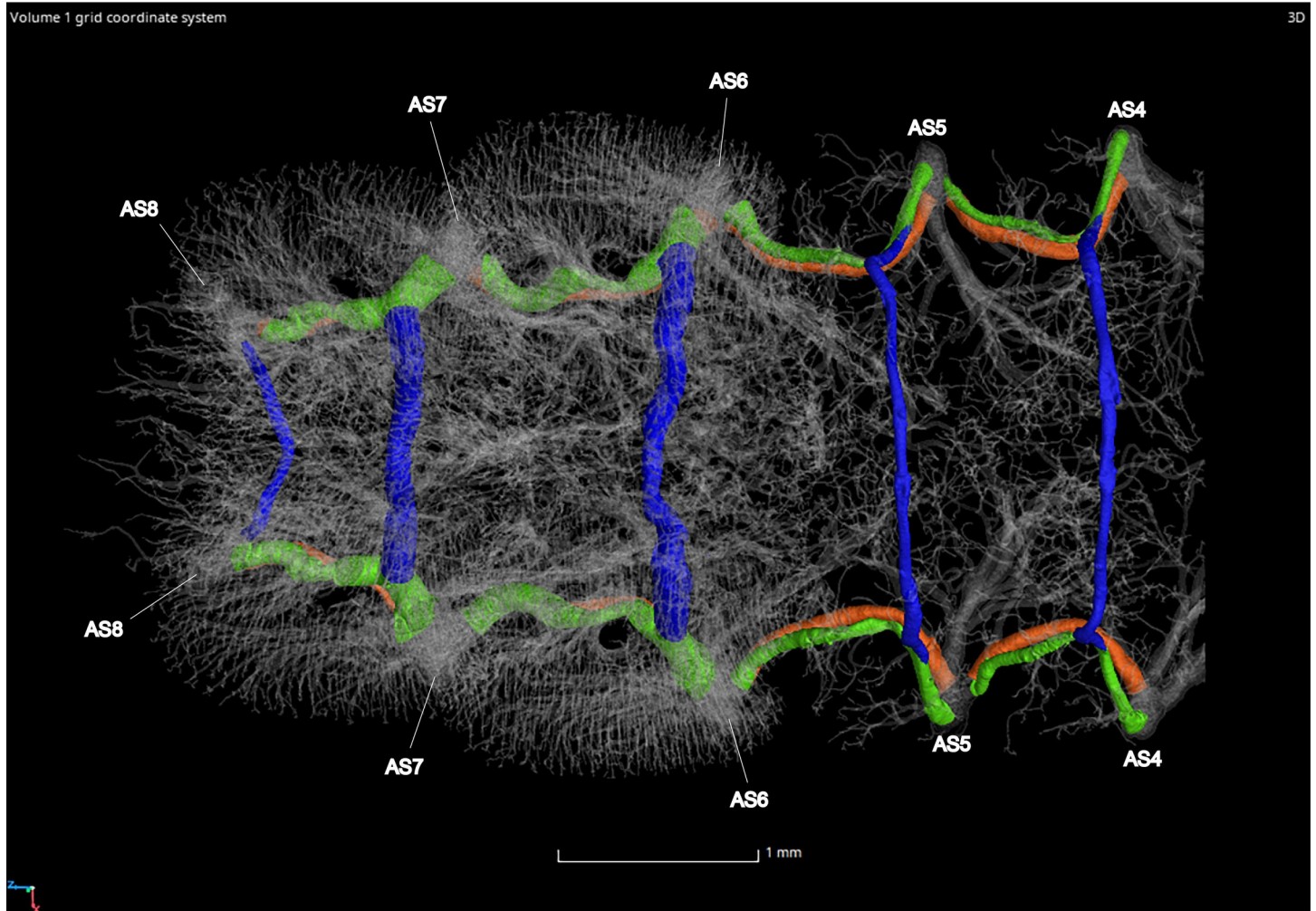

**Fig 24. Ventral view of the adult male *Photinus pyralis* tracheal system.** Ventral commissures (VC; blue), dorsal longitudinal trunks (DLT; orange), and ventral longitudinal trunks (VLT; green) highlighted for volumetric analysis.

radially branching into tracheal twigs, each twig culminating in a trifurcation (or perhaps a bifurcation in some cases or taxa) of tracheoles at the photocytes (Fig 31A–31D) [32–34]. In the adult male *P. pyralis*, the volume of the tracheal brush in segments VI and VII combined was 0.26 mm$^3$, which makes up 2.29% of the total volume in those segments (Tables 3 and 5). Comparing this value to the total tracheal volume percent in those segments (7.31% including the brush), we see that the tracheal brush in the adult male contributes a sizable amount of volume to the tracheal system in luminous segments. In the adult female, the tracheal brush was 0.02 mm$^3$, which makes up 0.47% of the total volume in segment VI. Comparing this value to the total tracheal volume in this segment (3.14%), the increase in volume seen in the female tracheal brush is not nearly as high as that present in the male. This difference was expected due to the smaller size of the female light organ. To compare the density of tracheae in the brush, a 0.7mm$^2$ square was segmented from the tracheal brush in the center and right lower edge of segment VI in the male (Fig 32) and in the center of segment VI in the female to count the number of tracheae in these spaces. Results indicate that there are approximately 185 tracheae per 0.7mm$^2$ in the center and 182 tracheae per 0.7mm$^2$ on the edges of the male tracheal

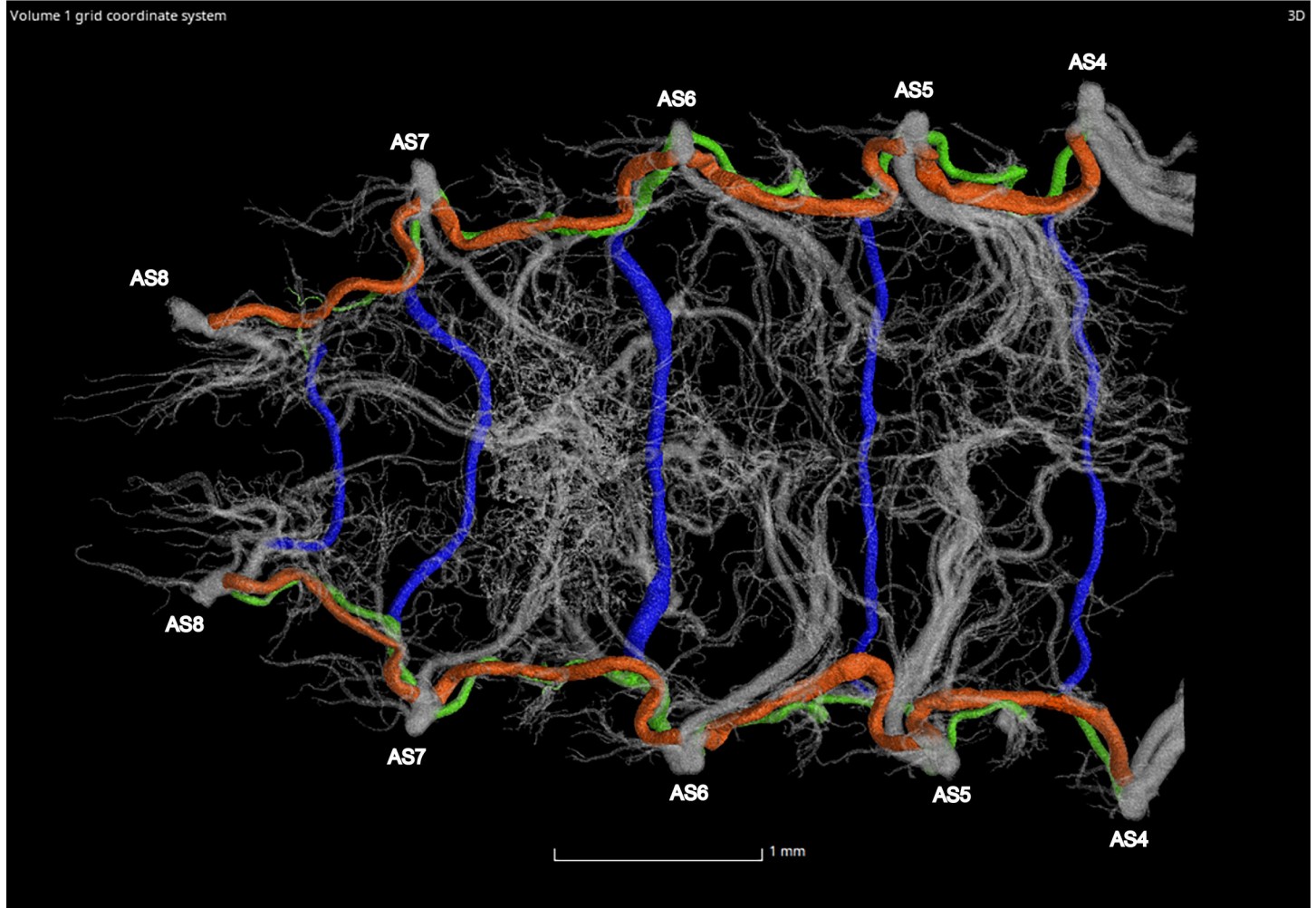

**Fig 25. Dorsal view of the adult female *Photinus pyralis* tracheal system.** Ventral commissures (VC; blue), dorsal longitudinal trunks (DLT; orange), and ventral longitudinal trunks (VLT; green) highlighted for volumetric analysis.

brush in segment VI. In the female, there are approximately 156 tracheae per 0.7mm$^2$ in the center of the tracheal brush.

## Discussion

### Abdominal spiracles

The abdominal spiracles of the *P. pyralis* larva and pupa are bifurous and have a filter apparatus (Figs 4A–4D, 8A and 8B), features that are correlated with their terrestrial lifestyle. Larval *P. pyralis* are typically found burrowing in the soil or on the soil surface after heavy rains, and larvae pupate in enclosed earthen cells (M. Branham pers. observ.). Filters at the openings of the tracheal system likely serve as a barrier for soil particles or other debris found in these habitats and prevent them from entering the tracheal system. could likely put the insect at risk if these particles were to enter the tracheal system. In contrast to the larva, adult *P. pyralis* are found either resting on foliage or flying in open areas such as fields dominated by low grass. The adult spiracles lack a filter apparatus and appear much different than that seen in the

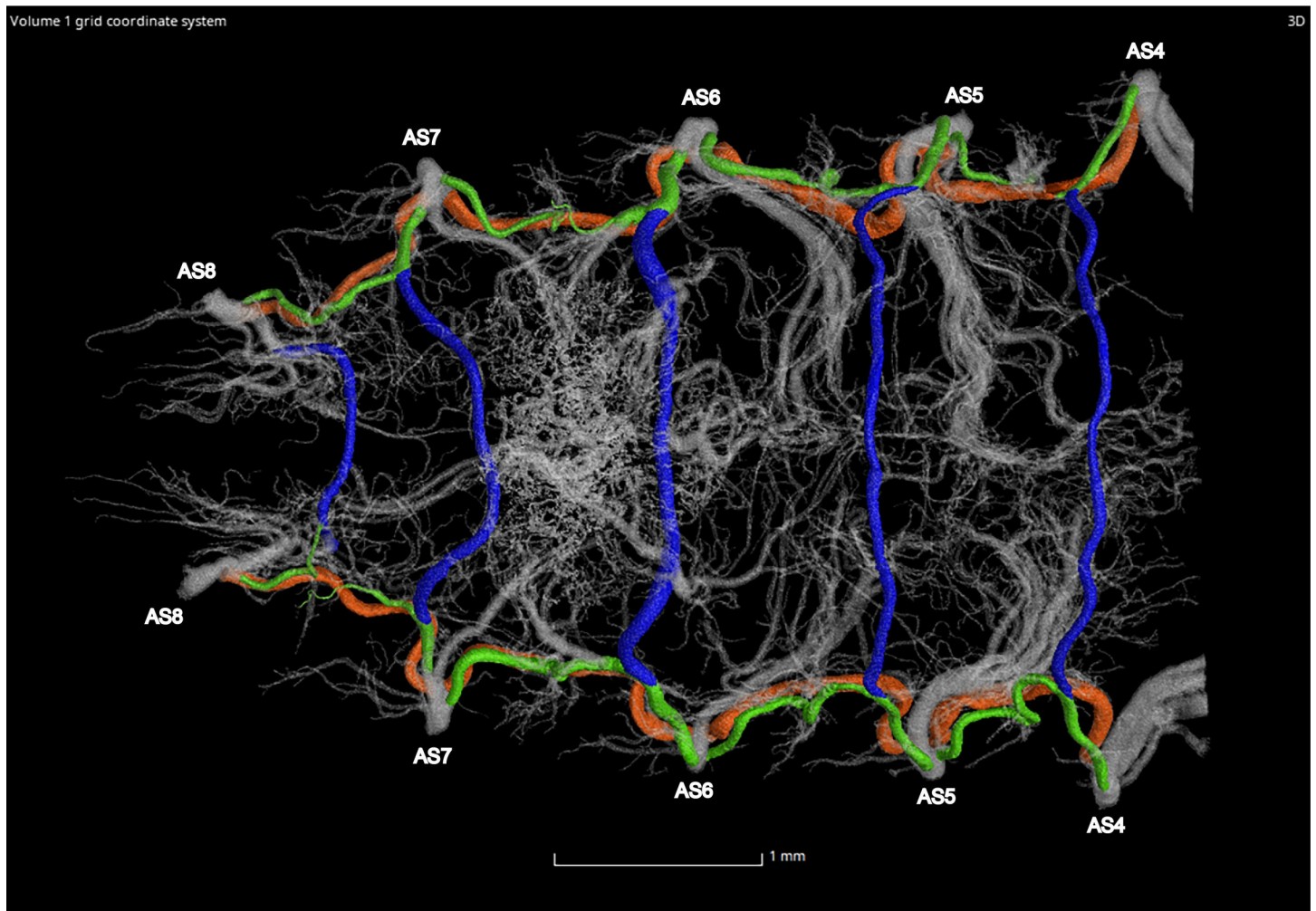

**Fig 26. Ventral view of the adult female *Photinus pyralis* tracheal.** Ventral commissures (VC; blue), dorsal longitudinal trunks (DLT; orange), and ventral longitudinal trunks (VLT; green) highlighted for volumetric analysis.

earlier life stages (Figs 15A–15E and 16A–16F). Adult *P. pyralis* would not face the same obstacles as terrestrial larvae and pupae, and therefore do not require the same spiracular structure.

The adult abdominal spiracles are uniform in size and structure, except for abdominal spiracle 1 (AS1) (Fig 15A, 15B, 15D and 15E). AS1 is much different from the other abdominal spiracles and resembles those of the thorax. It is significantly larger and lacks the levers seen in the remaining abdominal spiracles. Although this spiracle is located on abdominal segment I, it primarily functions to ventilate the thoracic tracheal system, specifically the flight musculature and hind wings [22]. AS2–8 are small and uniform in the adults of both sexes (Figs 15A, 15C and 16A–16F). The levers and occlusor muscle function as a closing mechanism specific to each spiracle (Figs 15C and 16A–16F).

Based on the significant proliferation of tracheae entering the light organ in adult *P. pyralis*, the oxygen demand and need for gas exchange within this organ is clearly significant. However, the size and shape of the spiracles on the abdominal segments that contain adult light organs are no larger than those on nonluminous segments. As all abdominal spiracles possess a closing mechanism, we hypothesize that these closing mechanisms are likely coordinated in

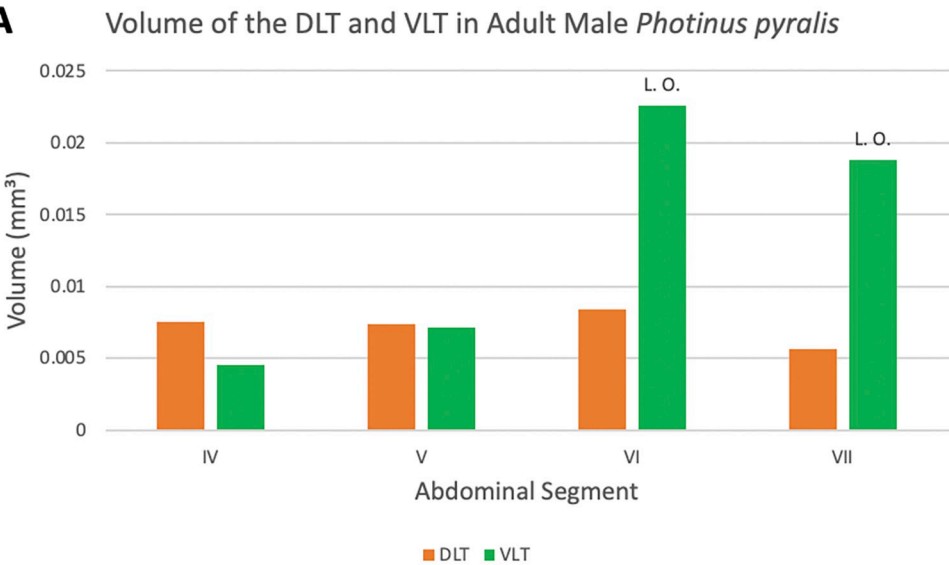

**Fig 27.** Volume of the dorsal longitudinal trunk (DLT) and ventral longitudinal trunk (VLT) in segments IV–VII in adult male (A) and female (B) *Photinus pyralis*. "L. O." indicates the presence of a light organ on that segment.

the context of abdominal contraction and expansion to facilitate ventilation of the abdomen and by extension, the light organs.

## Spiracle branches (SB)

The spiracle branch (SB) is found just inside of the spiracle past the atrium. In the larva and 3–4 day old pupa, the SB is long and thin (Figs 1–3 and 5–7). The SB is uniform in size in segments IV–VIII. In the adult male and female, the SB has become enlarged and short, functioning more as an atrium than a true branch (Figs 9–14 and 17–19). The SB of the male's luminous segments (VI and VII) appears enlarged when compared to the nonluminous segments. This specialization is not seen in the female's luminous segment (VI), though all SBs in the female have a structure resembling the adult male. Reducing the length of the SB in the

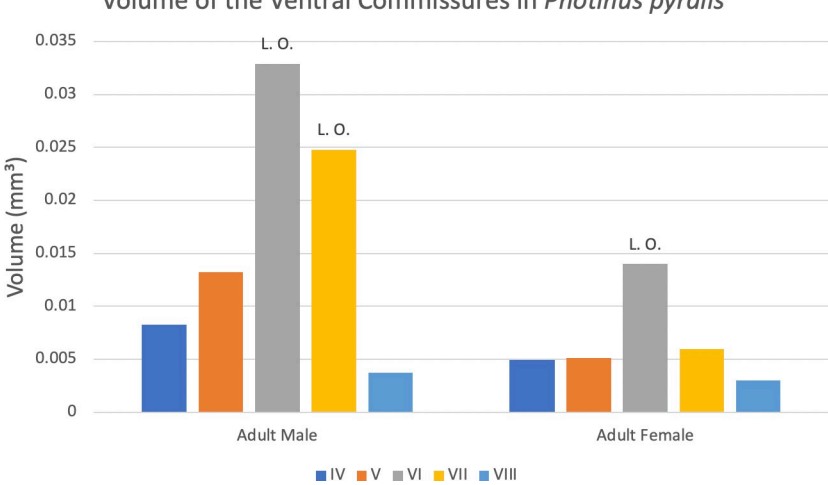

**Fig 28. Volume of the ventral commissures in abdominal segments IV–VIII of adult male and female *Photinus pyralis*.** "L. O." indicates the presence of a light organ on that segment.

**Fig 29. Ventral view of the adult male *Photinus pyralis* tracheal system with the tracheal brush highlighted in yellow (restricted to abdominal segments VI and VII).**

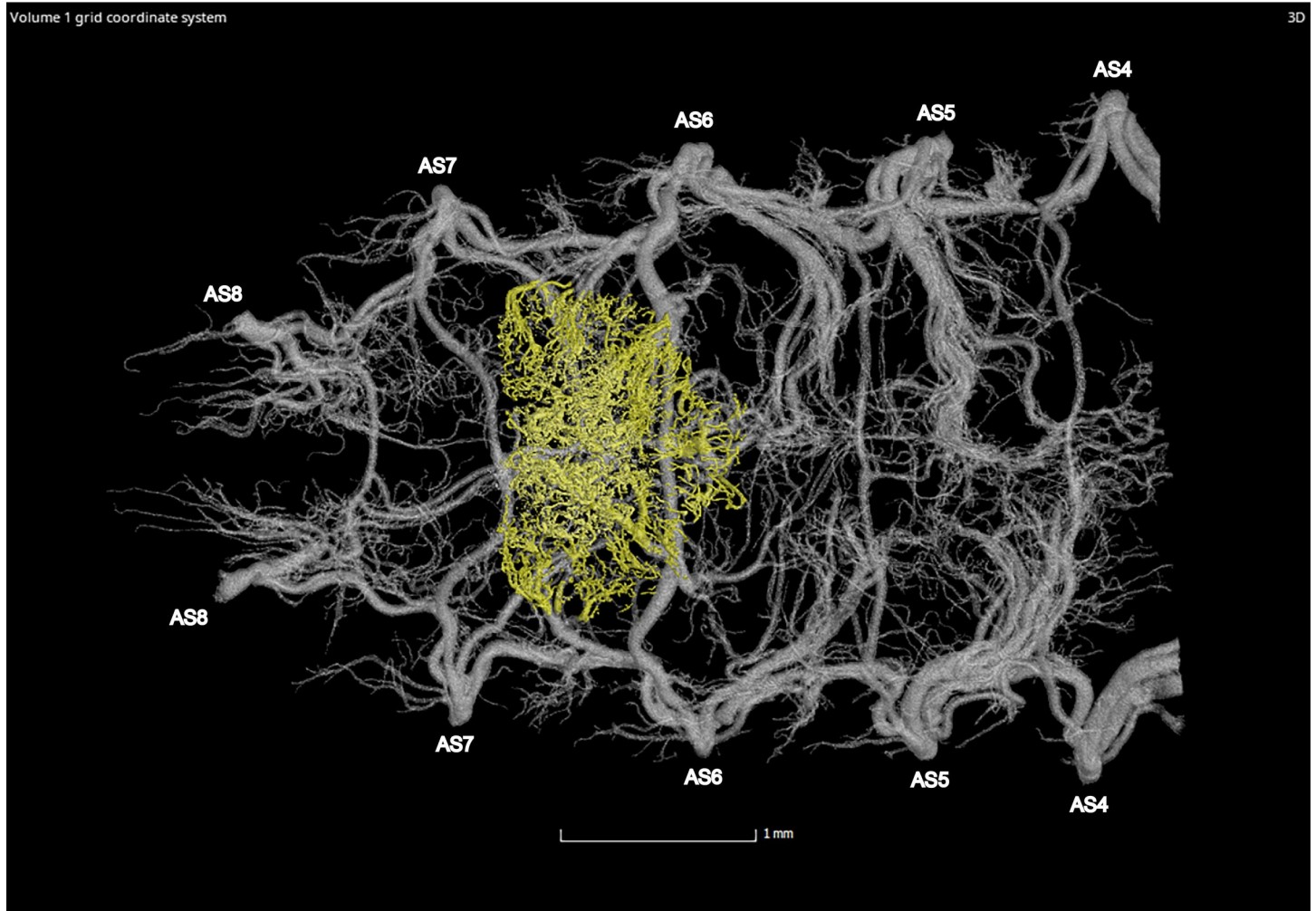

**Fig 30. Ventral view of the adult female *Photinus pyralis* tracheal system with the tracheal brush highlighted in yellow (restricted to abdominal segment VI).**

adult life stage would allow for greater speed of the movement of gases through the abdomen. It is expected that an increased rate of gas delivery through the tracheal system would be necessary to support a flashing light organ, where the presence and absence of different respiratory gases is essential in triggering the on/off function [34–36]. Interestingly, the structure of the spiracles creates a constriction at the entrance to the tracheal system (Fig 22). Apparently, this opening has not increased in size along with the SB in the adult life stage to permit a greater amount of gas exchange during respiration. This lack of spiracular modification alongside the SB contrasts to the enlarged thoracic and first abdominal spiracles feeding the metabolically demanding flight muscles in beetles, as well as modifications in other insects such as the enlarged eighth abdominal spiracles feeding the terminal tracheal tufts and functional lung in many lepidopteran larvae [37]. There are several potential reasons for this, including evolutionary and developmental constraints, risks associated with larger spiracle openings (larger environmental particles entering the tracheal system), and functional limitations. Spiracles may open and close asynchronously to direct oxygen flow into and out of certain abdominal segments [38] and could be more efficient at their present size despite the lack of modification to the luminous segments.

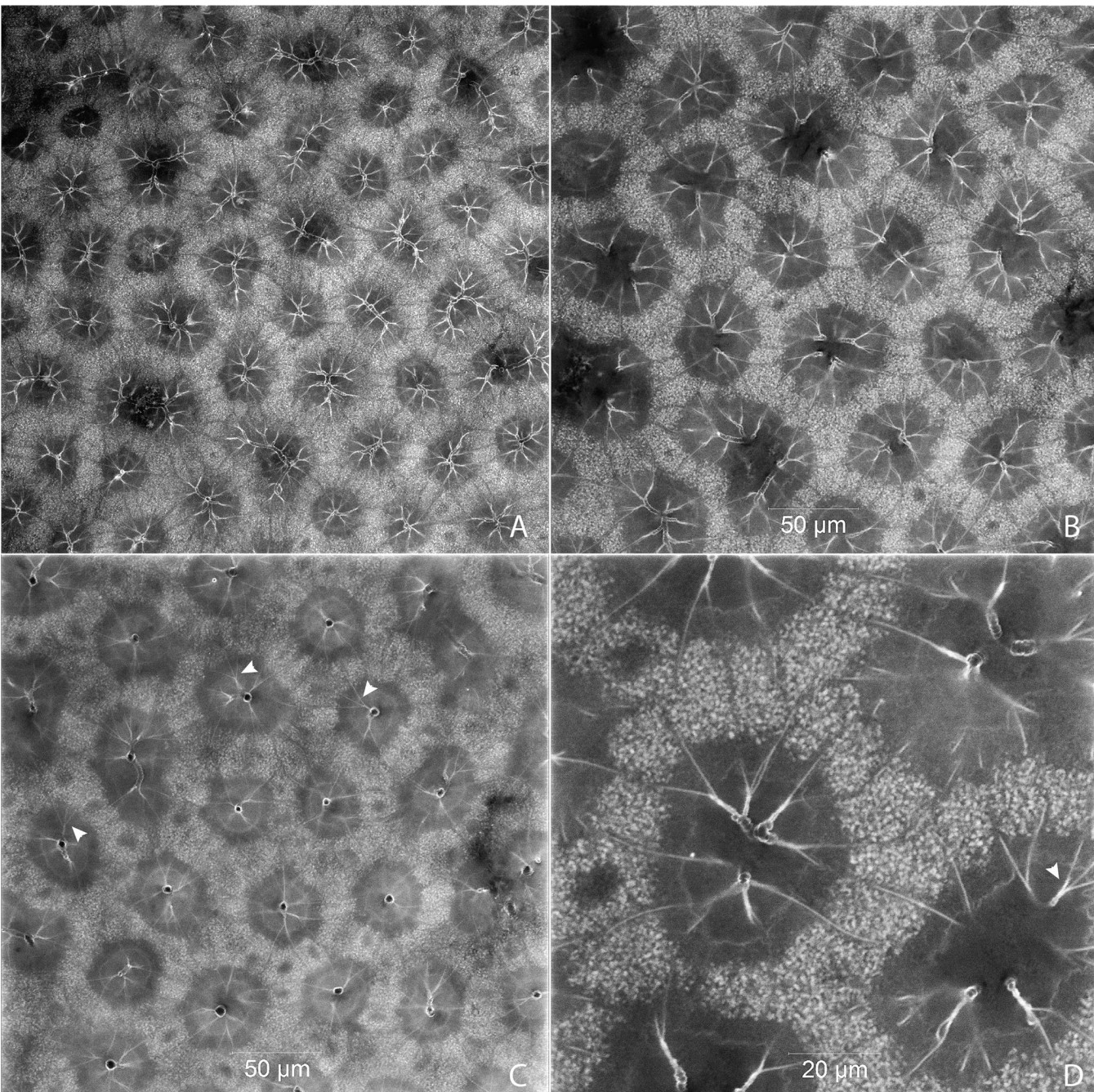

**Fig 31. Tracheae and tracheoles in each tracheal end organ cylinder comprising the adult male tracheal brush in *Photinus pyralis*.** Preparation was cleared using lactic acid. Arrowheads pointing to junctions where tracheal twigs branch into three tracheoles. A, B, and C show terminal tracheae near the ventral surface of the photogenic layer; C shows an optical section further from the ventral surface.

### Dorsal longitudinal trunk (DLT) and ventral longitudinal trunk (VLT)

In all life stages, the DLT forms a complete connection between SBs in abdominal segments IV–VIII (Figs 1–3, 5–7, 9–14, 17–19, 23–26). The DLT is uniform in size across segments and does not appear to show any major modifications in luminous segments (Figs 23–26, 27A, 27B; Table 4). Alternatively, there is considerable modification in the VLT between life stages

**Table 5. Volume of the tracheal brush in adult male and female *Photinus pyralis*.**

| | Tracheal Brush Volume (mm$^3$) | Total Tracheal Volume (mm$^3$) |
|---|---|---|
| **Adult Male (Segments VI+VII)** | 0.2604 | 0.8284 |
| **Adult Female (Segment VI)** | 0.0162 | 0.1355 |

Total tracheal volume is the measure of abdominal tracheation in the whole segment(s) including the tracheal brush.

and sexes. In the larva and pupa, the VLT branches from the SB and ends in visceral tracheae. The connection of the VLT is incomplete between the SBs, leaving the DLT as the only complete connection of the spiracles between SBs. The VLT are uniform in size across these two life stages and do not appear to have any modification in the bioluminescent active segment VIII. In adults, the VLT has a much different structure. The VLT of the adult male forms a complete connection between SBs in segments IV–VIII (Figs 23 and 24). The VLT also increases significantly in volume in the luminous segments (VI and VII) (Fig 27). This is expected due to the increased needs for respiratory gases in the ventral portion of the body

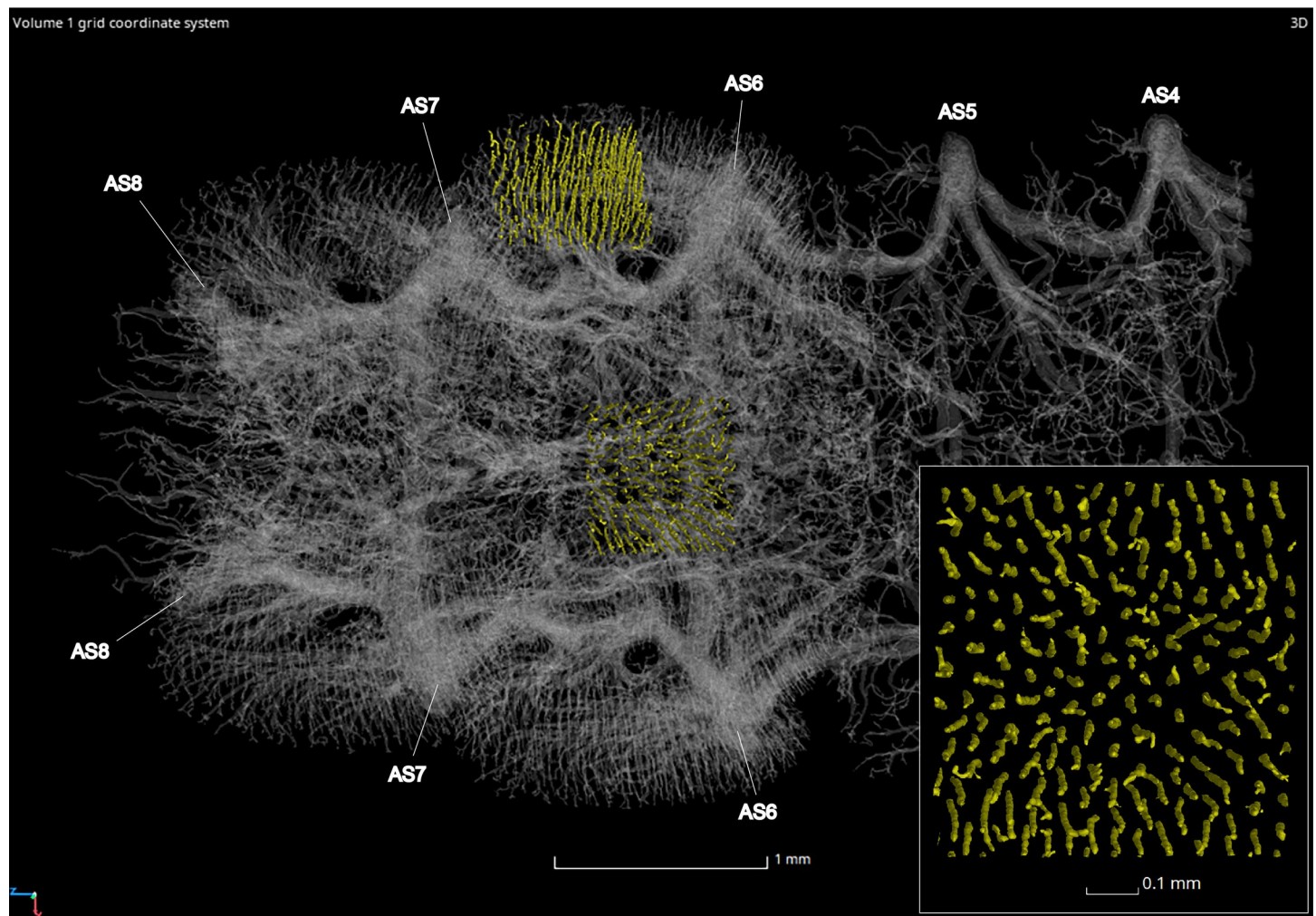

**Fig 32. Ventral view of the adult male *Photinus pyralis* abdominal tracheal system showing segmentation of two 0.7mm$^2$ sections of the tracheal brush (highlighted in yellow) to assess tracheal density.** Inset image highlights the organization of the tracheae penetrating the center of the light organ on abdominal segment VI.

where the light organ is located. However, the development of a complete VLT connection cannot be attributed to the presence of the light organ due to this connection's occurrence in both luminous and nonluminous abdominal segments, as well as its presence in other beetle families. The VLT becomes peculiar in the adult female (Figs 25 and 26). Complete connections of the VLT between SB occur in both sides of segment V, but only in the left side of segment VI and the right side of segment VII. When the VLT is complete the trunk is uniform in diameter, whereas an incomplete VLT gradually tapers into smaller visceral tracheae. This makes it relatively simple to determine if an incomplete connection is naturally occurring or if the VLT became fluid-filled prior to scanning. The VLT of the right side of segment IV was partially filled with fluid prior to the scan and therefore was not completely visible. Although we cannot determine if this connection was complete or incomplete, it is interesting that the center of the VLT connection appears to have become fluid-filled. This may indicate that the connection was incomplete, providing access for fluids to enter the VLT without obstructing the view of the connections to the SB, though these fluids may also have entered this portion of the VLT through visceral tracheae branching from the VLT. The reason for these incomplete connections is unknown. We are unsure if this is due to developmental anomalies in this individual or if these incomplete connections will be present in other female *P. pyralis*. The volume of the VLT was found to increase in the luminous segment (VI) though not to the significant extent seen in the male's luminous segments (Table 4; Fig 27). The increase in the VLT in adult light organ segments is reported here for the first time. The increase in volume in this tracheal component clearly indicates its potential importance in the increased delivery of respiratory gases to the light organ.

## Ventral commissure (VC)

The VC branches from the VLT and forms a complete connection between two VLT in all abdominal segments of the larva, pupa, adult male, and adult female. In the larva and pupa, the VC is uniform in size across segments V–VIII (Figs 1–3 and 5–7). In the adult male, the VC is considerably larger in the luminous segments VI and VII than the nonluminous segments (Figs 23, 24 and 28; Table 4). The same is true for the female, where the VC of luminous segment VI is larger than those of the nonluminous segments (Figs 25, 26 and 28; Table 4). The increase in size of the VC in luminous segments is correlated with the presence of the flashing light organ. In the larva and pupa, we see no clear difference in size in segment VIII where the functional larval light organ is present. In adults, we see an increase in luminous segments specifically. The VC branches into much of the tracheation that supplies the light organ tissues with respiratory gases. In flashing species, it appears logical that the commissure increased in volume to support the needs of the light organ.

## Tracheal brush

In the light organs of both sexes of adult *P. pyralis*, a high concentration of tracheae run through the light organ tissues and are confined to the area of the abdomen that contains the light organs (Figs 29 and 30). We propose the name "tracheal brush" for this structure, as its numerous parallel terminal tracheae, which penetrate the light organ tissues, resemble the bristles of a brush. This peculiar arrangement of the terminal tracheae, having linear tubes extending through the tissue with smaller branches/twigs radiating from the central tracheal column, is perhaps similar to the central tracheal columns extending through the thoracic flight muscle bundles in Odonata (S. Davis pers. observ.). As alluded to in the introduction, the increased metabolic demands in pterygote thoracic flight muscles and flashing lampyrid light organs appear to have culminated in similar increased tracheal arborization patterns. This tracheal

brush has been noted in other firefly species and appears correlated with a flashing light organ [19, 21]. In female *P. pyralis*, the tracheal brush is composed of tracheae branching from the VC and VLT of segment VI. In males, the tracheal brush is composed of tracheae branching from the VC, VLT, and VVi of segments VI and VII. The addition of tracheae branching from the VVi in the tracheal brush allows the light organ to extend to the lateral edges of the luminous segments. The tracheal brush contributes a considerable amount of volume to the tracheal system of luminous segments (Table 5). In *P. pyralis*, the tracheal brush of the adult male contributes a greater amount of volume to the tracheal system than that in the female (see Tracheal Brush, Results section). This is expected due to the presence of the light organ covering the entire ventral surface of two segments in the male (VI and VII) and only occurring on the central ⅓ of the ventral surface of one segment (VI) in the female.

To determine if the differences in the tracheal brush between sexes were strictly due to the size of the light organ, we tested the density of tracheae making up the brush in segment VI in the adult male and female. Additionally, we segmented and counted tracheae in two areas of the male tracheal brush, the center and right lower edge, to determine if there was uniform density of tracheae across the male tracheal brush. The male tracheal brush has a density of approximately 185 tracheae per $0.7mm^2$ in the center of segment VI and 182 tracheae per $0.7 mm^2$ on the lower right edge of segment VI. The difference of 3 tracheae could be explained by the plane in which the square was measured, where tracheae along the edge may have been counted in the center measurement, but just outside of the field of view in the edge measurement. Therefore, based on these two measurements it would appear that the density of tracheae through the male tracheal brush is uniform. When comparing these values to the female, where there are approximately 156 tracheae per $0.7mm^2$ in the center of the tracheal brush, the results indicate that the tracheal brush of the male has a higher density of tracheae than that of the female. We report here for the first time that the male tracheal brush in *P. pyralis* has both greater density and volume of tracheae than that of the female.

## Movement of gases through the tracheal system

The realization that insect respiration is quite active (a departure from the more than century-old assumption of passive diffusion) alludes to the possibility of various modes of tracheal ventilation that may be employed with the advent of flashing bioluminescence. As the abdomen is highly flexible in fireflies, active compression or pumping in this region, a mode of ventilation common among insects, could work in concert with spiracular operation or other forms of active ventilation [39, 40]. Despite earlier observations that abdominal ventilation may be minimal in many beetles [41, 42], at least fireflies warrant critical examination of such hypotheses particularly as the movement of respiratory gases appears vital to the control of flashing [34–36]. While few taxa have been examined, various patterns of airflow through the insect body have been described involving combinations of spiracular controls, tracheal and air sac valves, and muscular contractions acting directly on tracheae/air sacs and modifying hemocoelic pressure, often compartmentally [43–47].

Although fireflies appear to lack air sacs, all of their spiracles (including thoracic) bear occlusor muscles and there even may be abdominal compartments that function in the above-mentioned fashion, possibly in concert with muscles such as the dorso-ventral pairs that are situated medio-laterally along the abdomen. Associated with the modes by which fireflies may ventilate their light organs, the respiratory pattern by which this occurs may be variable as well. Patterns that have been characterized in many insect orders include continuous, cyclic, and discontinuous gas exchange [47]. Explanations have been developed for why, when, and where in insect phylogeny these various modes occur but appear to be labile based on

environmental factors and metabolic activity [48, 49]. It would be curious to examine what patterns exist in fireflies and the relation of these patterns to a metabolically demanding light organ that has been under intense sexual selection. Additionally, hemoglobin occurrence in lampyrids has not been discussed with regards to the operation or efficiency of the light organ. Given that intracellular hemoglobins are present and expressed in all insects including lampyrids [50], it would be fascinating to see if their expression is correlated with light organ presence to mediate gas exchange among photogenic cells and tracheoles and among photogenic cell organelles. There may be elevated levels of hemoglobins in light organ tissues and perhaps it may even be regulated spatially and to particular transcripts.

## Correlation between emission type and complexity of the tracheal system

Fireflies exhibit a variety of bioluminescent signalling types from glowing to flashing [2]. Because those with flashing bioluminescent signals require a more sophisticated mechanism for producing light, and due to the increase in respiratory gases required for such a signal, the emission type of bioluminescence appears correlated with the morphological specialization of the tracheal system [21]. We see this correlation in *P. pyralis*, where the glowing larvae have a simple tracheal network compared to that of the flashing adults (Figs 1–3, 5–7, 9–14, 17–19). The differences observed in our study support this correlation between signal type and complexity, where there is increased specialization of the tracheal system in subsequent life stages, and further specialization when comparing the adult male and female. Furthermore, differences are seen in the tracheal system of the luminous segments between sexes, including branching patterns, density, and volume, which represents the differing respiratory specializations of adult bioluminescence.

Apart from structural changes across life stages, an increase in volume of the tracheal system occurs as well. In both adult male and adult female, we see a large increase in tracheal volume when progressing from pupa to adult (Table 3). However, the increase from pupa to adult male is much larger than that of pupa to adult female. A larger increase from pupa to adult male is expected due to the difference in respiratory demands of the light organ between the two sexes. The results of the volumetric analyses show that there are larger tracheal features in the luminous segments of both sexes (Figs 23–26; Table 4). The volume of the tracheal system in luminous segments relative to the size of the segment is considerably higher in the adult male, but this same level of increase is not seen in the female (Table 3). While the total volume of segment VI did increase when compared to segment V, the volume of the tracheal system only increased $0.001\text{mm}^3$ between the two segments (Table 3). Enlargement of segment VI is expected due to the presence of the adult light organ, and an increase in tracheal volume was expected to accompany this change with the additional volume from the enlarged VC, VLTs, and tracheal brush. We did not find the expected overall increase in the female relative to segment size as seen in the male despite both having larger features in their luminous segments. This may be due in part to the smaller size of the female light organ, though it also reflects the difference in specialization between sexes where the male's features (VLT and VC) have become much more enlarged when compared to the female and increase overall volume as a result.

## Selective pressures likely driving increasing complexity of the tracheal system across life stages

The flashes produced by male *P. pyralis* appear to function within both a sexual selection and natural selection context [51]. Male *P. pyralis* fireflies are commonly found flashing in very high numbers flying over open fields. Based on these high densities, male-male competition is

intense, with only a percentage of males receiving a female response. In this species, females preferentially respond to males that produce flashes of a greater intensity than their rivals [14]. While clearly functioning as a courtship signal, male *P. pyralis* flashes can also be used as an aposematic signal directed to potential predators [16, 17]. Bats are the primary aerial predators in the same habitats and same activity periods as *P. pyralis* males are flashing. Leavell et al. [17] demonstrated that the big brown bat, *Eptesicus fuscus* [52], which has overlapping distributions with *P. pyralis*, can learn to recognize and then avoid these fireflies using information received either as a sonar (acoustical) signature or the bioluminescent (visual) flash. These findings suggest that like their bioluminescent flashes, the adult light organs in *P. pyralis* have likely been shaped by selective forces consistent with both sexual selection and natural selection. Increased levels of photic output likely requires and benefits from high levels of tracheation that support these flashing light organs.

## Conclusions

Differences in the tracheal system in subsequent life stages of holometabolous insects are expected. What makes this firefly's tracheal modifications unique is the position of the light organs and thus the concentration of differences in the ventral portion of the body and in the luminous abdominal segments of the abdomen in adults. This makes fireflies an ideal organism for studying the tracheal system and its modifications to support complex, novel characters (such as the adult light organ). The tracheal system of *P. pyralis* changes across life stages, most notably in abdominal segments that develop an adult light organ. In the luminous segments of the adult male and female, several tracheal features have become enlarged, which is evidence of the increased respiratory requirements of a flashing light organ. Volumetric analyses and selective segmentation revealed an increase in some of the tracheal features of the luminous segment in the female (VC and VLT in segment VI) where an overall increase in tracheal volume relative to segment volume was not observed between nonluminous and luminous segments. An increase in volume of features of the tracheal system was seen in the adult male (VC and VLT in segments VI and VII) as well as an overall increase in tracheal volume in the luminous segments. The differences between sexes can be explained due to the behaviors exhibited during courtship signalling, where females are more selective of who they flash to, their flashed responses aimed at a specific male, while males produce high intensity flashes, visible over greater distances to advertise to females. Additionally, serious consideration should be given to the potential control of airflow by the insect when investigating the firefly light organs. If fireflies are opening and closing spiracles asynchronously and moving respiratory gases into and out of certain parts of the body, the physical mechanics of flashing the light organ may be even more involved than previously thought.

Bioluminescent emission type across life stages appears correlated with the elaboration of the tracheal system in *P. pyralis*. Glowing larvae and pupa have a simple tapering tracheal system while adults of the same species have enlarged tracheal trunks, branches, and commissures, and a concentration of tracheae making up the tracheal brush. While this is evident in the results of this study and in previously published literature, a more comprehensive study will need to be done across multiple taxa to confidently conclude that emission type is correlated with the elaborate tracheal modifications described herein. Multiple organ systems are likely modified in the abdomen of flashing fireflies to allow both the occurrence and function of the light organ. This detailed study of the tracheal system across life stages of this firefly illuminates which structural components have become modified in the adult stage and are either incorporated into or appear to support the function of flashing light organs.

## Acknowledgments

We would like to thank Dr. Edward Stanley for assistance with 3D segmentation and the use of resources in the Digital Imaging Division at the Florida Museum of Natural History, Dr. Gary Scheiffele from the University of Florida Research Service Centers for training and assistance with the micro-CT scanner, and Mr. and Mrs. Steven and Patricia Dunn for assistance with collecting specimens. Additionally, we would like to thank Dr. Elaine Seaver, Dr. Peter DiGennaro, and Dr. Paul Skelley for their comments on a draft of this manuscript.

## Author Contributions

**Conceptualization:** Kristin N. Dunn, Steven R. Davis, Seth M. Bybee, Marc A. Branham.

**Data curation:** Kristin N. Dunn, Steven R. Davis.

**Formal analysis:** Kristin N. Dunn, Steven R. Davis, Hollister W. Herhold.

**Funding acquisition:** Kathrin F. Stanger-Hall, Seth M. Bybee, Marc A. Branham.

**Investigation:** Kristin N. Dunn, Steven R. Davis.

**Methodology:** Kristin N. Dunn, Steven R. Davis, Hollister W. Herhold.

**Project administration:** Marc A. Branham.

**Resources:** Steven R. Davis, Marc A. Branham.

**Supervision:** Marc A. Branham.

**Validation:** Kristin N. Dunn, Steven R. Davis, Hollister W. Herhold.

**Visualization:** Kristin N. Dunn, Steven R. Davis.

**Writing – original draft:** Kristin N. Dunn, Steven R. Davis.

**Writing – review & editing:** Kristin N. Dunn, Steven R. Davis, Hollister W. Herhold, Kathrin F. Stanger-Hall, Seth M. Bybee, Marc A. Branham.

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
