## [Decision Letter · Decision Letter 0]

25 Feb 2022

PONE-D-21-39869Morphological changes in the tracheal system associated with light organs of the firefly *Photinus pyralis* (Coleoptera: Lampyridae) across life stagesPLOS ONE

Dear Dr. Dunn,

Thank you for submitting your manuscript to PLOS ONE. After careful consideration, we feel that it has merit but does not fully meet PLOS ONE’s publication criteria as it currently stands. Therefore, we invite you to submit a revised version of the manuscript that addresses the points raised during the review process.

We look forward to receiving your revised manuscript.

Kind regards,

Feng ZHANG, Ph.D.

Academic Editor

PLOS ONE

Journal Requirements:

Reviewers' comments:

Reviewer's Responses to Questions

**Comments to the Author**

1. Is the manuscript technically sound, and do the data support the conclusions?

Reviewer #1: Yes

2. Has the statistical analysis been performed appropriately and rigorously? 

Reviewer #1: Yes

3. Have the authors made all data underlying the findings in their manuscript fully available?

Reviewer #1: Yes

4. Is the manuscript presented in an intelligible fashion and written in standard English?

Reviewer #1: Yes

5. Review Comments to the Author

Reviewer #1: Fireflies are fascinating beetles attracted extensive attention from evolutionary biologists, entomologists, and the public in general. Many species of fireflies, as both adults and larvae, are able to produce lights for defense, communications and other proposes, so oxygen is important of for firefly’s bioluminescent chemical reaction. There is no doubt that the development of the tracheal system and its modification are key to understanding this issue. The authors employ micro-CT scanning to reveal the morphological comparisons across the larva, pupa, and adult life stages, revealing invaluable information for understanding character evolution in a firefly. This work is nicely presented; the illustrations and figures are fantastic; and the conclusions are supported by the given data. I urge to publish this article after a very minor revision.

1. Format: many ‘-’ should be replaced by en dash, which means ‘to’.

2. Data availability: The authors need to deposit the original raw micro-CT slice into online depository. Also, it is better to present 3D videos of the tracheal system of different stages as Supplemental Data.

6. PLOS authors have the option to publish the peer review history of their article (what does this mean?). If published, this will include your full peer review and any attached files.

Reviewer #1: No

---

## [Author Response · Author response to Decision Letter 0]

22 Apr 2022

Kristin N. Dunn

University of Florida

1881 Natural Area Drive

Gainesville, FL 32611

April 8, 2021

Dr. Feng Zhang

Academic Editor

PLOS ONE

Dear Dr. Zhang,

This letter is in response to the review of an original research article titled “Morphological changes in the tracheal system of the firefly Photinus pyralis (Coleoptera: Lampyridae) across life stages” submitted for publication consideration in PLOS ONE. 

Addressing the points raised during the edit and review: 

File Naming – File names have been checked.

Grant Information – The correct funding information is listed below. 

Funding was provided by the National Science Foundation (https://www.nsf.gov/) M.A.B. DEB-1655936, a collaborative research grant with K.S.H. DEB-1655908 and S.M.B. DEB-1655981

Funding for the AMNH Micro-CT scanner was provided via an NSF instrumentation grant MRI-R2 0959384

H.W.H. was supported by a Richard Gilder Graduate School Fellowship

Data Availability – Raw micro-CT scan data have been deposited into Morphosource.org. Access links are below.

Project: https://www.morphosource.org/projects/000434376?locale=en

Larva tifs: https://www.morphosource.org/concern/media/000434523?locale=en

Pupa tifs: https://www.morphosource.org/concern/media/000434527?locale=en

Adult Male tifs: https://www.morphosource.org/concern/media/000434514?locale=en

Adult Female tifs: https://www.morphosource.org/concern/media/000434531?locale=en

Reference List – References remain the same as in the initial submission, with two exceptions: Citations 41-43 were missing from the in text citations. This has been corrected (Page 31). Two citations, 34 and 35, were reordered based on citation date within text citation (Page 40).

En dashes have been placed throughout the manuscript where appropriate. 

Additional minor changes:

Bold type removed from title (Page 1).

Spelling correction (Page 5).

Duplicate table captions deleted (Pages 15, 23).

Spelling correction (Page 16).

General grammar throughout.

Although our reviewer suggested videos as supplementary data, we feel that our figures accurately portray the evidence we discuss in the article and that further imaging is not needed. Due to the time requirement of creating such videos, we feel it is in our best interest to move forward with the imaging as is but do appreciate this suggestion and will take this into consideration in future works. 

Please address all correspondence concerning this manuscript to me at KristinDunn@ufl.edu

Sincerely,

Kristin N. Dunn, MS

Doctoral Candidate, Department of Entomology and Nematology

University of Florida

(757) 619-7094

kristindunn@ufl.edu

Co-Authors:

Dr. Steven Davis

Division of Invertebrate Zoology

American Museum of Natural History

(785) 424-0810 sdavis@amnh.org

Hollister Herhold

Division of Invertebrate Zoology

American Museum of Natural History

hherhold@amnh.org

Dr. Kathrin Stanger-Hall

Department of Plant Biology

University of Georgia

(707) 542-1870

ksh@uga.edu

Dr. Seth M. Bybee

Department of Biology

Brigham Young University

(801) 422-3152

seth.bybee@byu.edu

Dr. Marc Branham

Department of Entomology and Nematology

University of Florida

(352) 273-3915 marcbran@ufl.edu

---

## [Editor Report · Decision Letter 1]

25 Apr 2022

Morphological changes in the tracheal system associated with light organs of the firefly *Photinus pyralis* (Coleoptera: Lampyridae) across life stages

PONE-D-21-39869R1

Dear Dr. Dunn,

We’re pleased to inform you that your manuscript has been judged scientifically suitable for publication and will be formally accepted for publication once it meets all outstanding technical requirements.

Kind regards,

Feng ZHANG, Ph.D.

Academic Editor

PLOS ONE
---

## [Editor Report · Acceptance letter]

6 May 2022

PONE-D-21-39869R1 

Morphological changes in the tracheal system associated with light organs of the firefly *Photinus pyralis* (Coleoptera: Lampyridae) across life stages 

Dear Dr. Dunn:

I'm pleased to inform you that your manuscript has been deemed suitable for publication in PLOS ONE. Congratulations! Your manuscript is now with our production department. 

Kind regards, 

on behalf of

Dr. Feng ZHANG 

Academic Editor

PLOS ONE